# Information-Theoretic Discrete Diffusion

**Moongyu Jeon**[1]    **Sangwoo Shin**[1]    **Dongjae Jeon**[2]    **Albert No**[1]*
[1]Department of Artificial Intelligence, Yonsei University
[2]Department of Computer Science, Yonsei University

## Abstract

We present an information-theoretic framework for discrete diffusion models that yields principled estimators of log-likelihood using score-matching losses. Inspired by the I-MMSE identity for the Gaussian setup, we derive analogous results for the discrete setting. Specifically, we introduce the Information–Minimum Denoising Score Entropy (I-MDSE) relation, which links mutual information between data and its diffused version to the minimum denoising score entropy (DSE) loss. We extend this theory to masked diffusion and establish the Information–Minimum Denoising Cross-Entropy (I-MDCE) relation, connecting cross-entropy losses to mutual information in discrete masked processes. These results provide a time-integral decomposition of the log-likelihood of the data in terms of optimal score-based losses, showing that commonly used losses such as DSE and DCE are not merely variational bounds but tight and principled estimators of log-likelihood. The I-MDCE decomposition further enables practical extensions, including time-free formula, conditional likelihood estimation in prompt–response tasks, and coupled Monte Carlo estimation of likelihood ratios. Experiments on synthetic and real-world data confirm the accuracy, variance stability, and utility of our estimators. The code is publicly available at `https://github.com/Dongjae0324/infodis`.

## 1   Introduction

Diffusion models have emerged as a powerful framework for generative modeling, enabling state-of-the-art performance in continuous domains such as image and audio generation (Sohl-Dickstein et al., 2015; Ho et al., 2020; Chen et al., 2021; Kong et al., 2021; Saharia et al., 2022). Central to these models is the idea of gradually corrupting data through a forward noising process, and learning to reverse this corruption via score-based loss (Hyvärinen and Dayan, 2005; Vincent, 2011; Song and Ermon, 2019; Song et al., 2021a,b).

Recent works have extended diffusion models to discrete domains, proposing models designed for categorical data (Hoogeboom et al., 2021; Austin et al., 2021; Campbell et al., 2022; Meng et al., 2022; Sun et al., 2023; Lou et al., 2024; Sahoo et al., 2024; Shi et al., 2024). These models offer a promising alternative to traditional autoregressive approaches (Radford et al., 2018, 2019; Brown et al., 2020), particularly for sequence modeling tasks that involve text and other symbolic structures (Li et al., 2022; Nie et al., 2025).

Continuous diffusion models benefit from a well-established information-theoretic foundation (Kong et al., 2023, 2024). In the Gaussian setting, the I-MMSE identity (Guo et al., 2005; Venkat and Weissman, 2012) connects the mutual information between clean and noisy variables to the minimum mean squared error (MMSE), offering both theoretical insight and a basis for likelihood estimation. A pointwise generalization of this identity yields closed-form decompositions of the data log-likelihood in terms of estimation losses (Kong et al., 2023). However, the discrete case has not yet been investigated from an information-theoretic perspective.

---

*Correspondence to: Albert No `<albertno@yonsei.ac.kr>`.

39th Conference on Neural Information Processing Systems (NeurIPS 2025).

In this work, we extend these ideas to discrete diffusion models, developing an information-theoretic framework that rigorously characterizes the relationship between mutual information and score-based loss in discrete settings. We first establish the Information–Minimum Denoising Score Entropy (I-MDSE) identity, which connects mutual information decay in the forward process to the minimum of the denoising score entropy (DSE) loss. This leads to a closed-form decomposition of the negative log-likelihood (NLL) into a time-integral of the minimum DSE, showing that the DSE loss, previously viewed as a variational bound, actually constitutes a principled estimator for likelihood estimation.

We then turn to masked (absorbing) diffusion models, where we formulate the pointwise denoising cross-entropy (DCE) loss and prove its equivalence to the DSE loss under certain time reparameterization. Leveraging this equivalence, we derive the Information–Minimum Denoising Cross-Entropy (I-MDCE) identity, which mirrors the I-MDSE relation but in the masked setting. The I-MDCE identity provides a parallel time-integral decomposition of the NLL along the minimum DCE trajectory, revealing that the DCE loss, like DSE, serves as a theoretically grounded training objective that enables exact estimation of the data likelihood.

Building on this decomposition via I-MDCE, we derive a time-free reformulation of the log-likelihood, expressed as an expectation over randomly selected unmasked token subsets. This formulation enables efficient Monte Carlo estimation of the NLL without diffusion-time integration and extends naturally to conditional likelihoods of the form $p_0(\mathbf{x}^{\text{target}}|\mathbf{x}^{\text{context}})$, allowing estimation in structured generative settings such as prompt–response modeling. Furthermore, by coupling the sampling paths of two sequences, our framework provides a principled Monte Carlo estimator for likelihood ratios, achieving unbiased and low-variance estimation compared to independently sampled baselines.

We validate our framework through experiments on both synthetic and real-world datasets. First, using synthetic datasets with known ground-truth distributions, we show that our estimators accurately recover both unconditional and conditional log-likelihoods. Next, we verify the variance reduction properties of our time-free likelihood estimator and the coupled likelihood ratio estimator against their respective baselines. Finally, we demonstrate the practical utility of our approach through auditing experiments on real-world data, where conditional likelihood estimates detect out-of-distribution inputs and reveal distributional shifts in LLaDA (Nie et al., 2025). These results confirm that our information-theoretic framework not only offers theoretical insight but also enables accurate and interpretable likelihood estimation in discrete generative models.

## 2 Preliminaries

### 2.1 Discrete Diffusion Models and Score Matching

Given data $x_0 \sim p_0$, the forward diffusion process is modeled as a continuous-time Markov chain (CTMC), governed by a linear ODE (Anderson, 2012; Campbell et al., 2022):

$$\frac{dp_t}{dt} = Q_t p_t, \quad p_0 = p_{\text{data}} \tag{1}$$

where $Q_t \in \mathbb{R}^{N \times N}$ is the time-dependent transition rate matrix, with $N = |\mathcal{X}|$ possible states. As $t \to \infty$, the marginal $p_t$ converges to a stationary distribution $\pi$. For tractability, it is common to assume a factored form $Q_t = \sigma(t)Q$, where $Q$ is a fixed matrix and $\sigma(t)$ is a positive scalar function.

The reverse process is also governed by another CTMC, described by the following ODE (Kelly, 1980; Sun et al., 2023; Lou et al., 2024):

$$\frac{dp_{T-t}}{dt} = \overline{Q}_{T-t} p_{T-t}, \quad \overline{Q}_t(y, x) = \begin{cases} \frac{p_t(y)}{p_t(x)} Q_t(x, y) & \text{if } x \neq y, \\ -\sum_{\tilde{y} \neq x} \overline{Q}_t(\tilde{y}, x) & \text{if } x = y. \end{cases} \tag{2}$$

To simulate the reverse process, one typically initializes at $p_T^\theta = \pi$ and replaces the marginal ratio $\frac{p_t(y)}{p_t(x)}$ in Eq. (2) with a learned approximation, yielding a parameterized family $\{p_t^\theta\}_{t=T}^0$.

Early methods (Austin et al., 2021; Campbell et al., 2022) modeled the reverse conditional $p_{0|t}$ directly, but suffered from combinatorial scalability issues. Meng et al. (2022) instead used an $\ell^2$ regression loss to approximate the marginal ratio $\frac{p_t(y)}{p_t(x)}$, which proved unstable in practice.

Subsequently, Lou et al. (2024) introduced the *denoising score entropy (DSE)* loss, resolving these scalability and stability issues. Specifically, they use a score network $s^\theta : \mathcal{X} \times [0, T] \to \mathbb{R}^N$ to

estimate the marginal ratio, where each output $s^\theta(x, t)_y$ corresponds to $\frac{p_t(y)}{p_t(x)}$. The network is trained by minimizing the DSE loss, defined pointwise as:

$$\ell_{\text{DSE}}(x_0, x, t, s_t) := \sum_{y \neq x} Q_t(x, y) \left( s_t(x)_y - \frac{p_{t|0}(y|x_0)}{p_{t|0}(x|x_0)} \log s_t(x)_y + K\left(\frac{p_{t|0}(y|x_0)}{p_{t|0}(x|x_0)}\right) \right), \quad (3)$$

where $K(a) = a(\log a - 1)$.

This loss is minimized when the score network recovers the true score $s_t^\star$, where $s_t^\star(x)_y = \frac{p_t(y)}{p_t(x)}$:

$$s_t^\star = \arg\min_{s_t} \mathbb{E}_{p(x_0, x_t)} \left[\ell_{\text{DSE}}(x_0, x_t, t, s_t)\right]. \quad (4)$$

Aggregating this loss over time yields the time-integrated DSE training objective:

$$\mathcal{L}_{\text{DSE}}^T(x_0) := \int_0^T \mathbb{E}_{p_{t|0}(x_t|x_0)} \left[\ell_{\text{DSE}}(x_0, x_t, t, s_t^\theta)\right] dt,$$

where $s_t^\theta = s^\theta(\cdot, t)$ is the learned score at time $t$.

Importantly, this loss also serves as a variational upper bound on the negative log-likelihood (NLL) of the sample $x_0$ under the learned distribution:

$$-\log p_0^\theta(x_0) \leq \mathcal{L}_{\text{DSE}}^T(x_0) + D_{\text{KL}}\left(p_{T|0}(\cdot|x_0)\|\pi\right).$$

This dual role of the DSE loss, as both a score-matching loss and a variational bound, makes it a principled and practical training criterion for discrete diffusion models.

## 2.2 Masked Diffusion with Absorbing Transition Matrix

In practice, discrete diffusion models are defined over sequences $\mathbf{x} = x^1 x^2 \cdots x^L \in \mathcal{X}^L$. A major challenge in this setting is the intractability due to the exponential size of $Q_t \in \mathbb{R}^{N^L \times N^L}$.

To address this, previous work (Campbell et al., 2022; Lou et al., 2024) assumes that each token evolves independently under a shared rate matrix $Q_t^{\text{tok}} = \sigma(t)Q^{\text{tok}} \in \mathbb{R}^{N \times N}$. This assumption significantly reduces the complexity of the score network. Specifically, one only needs to estimate marginal ratios for sequence pairs that differ by a single token:

$$s^\theta(\mathbf{x}, t)_{i, \hat{x}^i} \approx \frac{p_t(x^1 \ldots \hat{x}^i \ldots x^L)}{p_t(x^1 \ldots x^i \ldots x^L)}, \quad \hat{x}^i \neq x^i.$$

The token-level forward transition is given analytically by $p_{t|0}(y^i|x^i) = \exp(\overline{\sigma}(t)Q^{\text{tok}})_{y^i, x^i}$, where $\overline{\sigma}(t) = \int_0^t \sigma(s)ds$. Closed-form expressions of the transition matrix $\exp(\overline{\sigma}(t)Q^{\text{tok}})$ are known only for specific choices of $Q^{\text{tok}}$, the uniform $Q^{\text{uniform}}$ and the absorbing $Q^{\text{absorb}}$ (Lou et al., 2024).

Of particular interest is the absorbing process, where $Q^{\text{absorb}}$ allows only transitions from unmasked tokens to a special mask token $[\mathbf{M}]$. This simplifies the reverse process by restricting the score computation to pairs $(\mathbf{x}, \hat{\mathbf{x}})$ that differ by exactly one masked position, with $\hat{x}^i \neq x^i = [\mathbf{M}]$.

A key property of absorbing diffusion is that the marginal ratios $\frac{p_t(\hat{\mathbf{x}})}{p_t(\mathbf{x})}$ admit *time-free reparameterization* (Ou et al., 2025). Specifically, for a pair $(\mathbf{x}, \hat{\mathbf{x}})$ with $\hat{x}^i \neq x^i = [\mathbf{M}]$ described above,

$$\frac{p_t(\hat{\mathbf{x}})}{p_t(\mathbf{x})} = \frac{e^{-\overline{\sigma}(t)}}{1 - e^{-\overline{\sigma}(t)}} p_0(\hat{x}^i|\mathbf{x}^{\text{UM}}), \quad (5)$$

where $\mathbf{x}^{\text{UM}}$ denotes the subsequence of unmasked tokens in $\mathbf{x}$.

This result motivates the use of a *time-independent* network $c^\theta : \{1, \ldots, N, [\mathbf{M}]\}^L \rightarrow \mathbb{R}^{L \times N}$ to predict the conditional distribution of unmasked tokens as

$$c^\theta(\mathbf{x})_{i, \hat{x}^i} = p_0^\theta(\hat{x}^i|\mathbf{x}^{\text{UM}}) \approx p_0(\hat{x}^i|\mathbf{x}^{\text{UM}}).$$

To simplify the time-integrated DSE loss, we reparameterize time $t$ using $\lambda(t) = 1 - e^{-\overline{\sigma}(t)}$, which monotonically increases from 0 to 1. In this coordinate system, Ou et al. (2025) introduced the *denoising cross-entropy (DCE)* loss:

$$\mathcal{L}_{\mathrm{DCE}}(\mathbf{x}_0) = \int_0^1 \frac{1}{\lambda} \, \mathbb{E}_{p_{\lambda|0}(\mathbf{x}_\lambda|\mathbf{x}_0)} \left[ \sum_{i=1}^L \mathbb{1}[x_\lambda^i = [\mathbf{M}]] \log \frac{1}{p_0^\theta(x_0^i|\mathbf{x}_\lambda^{\mathrm{UM}})} \right] d\lambda.$$

Importantly, they also proved that this loss is equivalent to the DSE loss in the full-noise limit:

$$\lim_{T \to \infty} \mathcal{L}_{\mathrm{DSE}}^T(\mathbf{x}_0) = \mathcal{L}_{\mathrm{DCE}}(\mathbf{x}_0) \tag{6}$$

which provides a simpler yet equally principled alternative to score matching. This formulation underlies recent large-scale masked diffusion language models such as LLaDA (Nie et al., 2025).

## 2.3 Information-Theoretic Diffusion

To motivate the development of information-theoretic tools for discrete diffusion, we begin by reviewing key results in the continuous setting, particularly the connection between mutual information and estimation error in Gaussian diffusion models.

Consider the standard Gaussian diffusion forward process (Sohl-Dickstein et al., 2015; Ho et al., 2020), where a data point $\mathbf{X} \sim p_0$ is corrupted via the noise channel

$$\mathbf{Z}_\gamma = \sqrt{\gamma} \, \mathbf{X} + \boldsymbol{\epsilon}, \quad \boldsymbol{\epsilon} \sim \mathcal{N}(0, I)$$

where $\gamma$ is the signal-to-noise ratio (SNR). This channel defines a reparameterized version of the diffusion forward process, often adopted in variational diffusion models (Kingma et al., 2021).

To reverse the diffusion, score-based models (Song and Ermon, 2019; Song et al., 2021b) learn the score function $\nabla \log p(\mathbf{Z}_\gamma)$, often reparameterized using Tweedie's formula (Efron, 2011) in terms of a denoiser $\hat{\mathbf{X}}_\theta$. The training objective can then becomes a denoising mean squared error (MSE) loss:

$$\mathcal{L}_{\mathrm{MSE}} = \frac{1}{2} \int_0^\infty \mathbb{E} \left[ \|\mathbf{X} - \hat{\mathbf{X}}_\theta(\mathbf{Z}_\gamma, \gamma)\|^2 \right] d\gamma.$$

which encourages the denoiser to approximate the MSE-optimal predictor $\mathbb{E}[\mathbf{X}|\mathbf{Z}_\gamma]$.

This denoising interpretation links naturally to a foundational identity in information theory: the I-MMSE relation (Guo et al., 2005), which states

$$\frac{d}{d\gamma} I(\mathbf{X}; \mathbf{Z}_\gamma) = \frac{1}{2} \mathrm{mmse}(\gamma),$$

where $\mathrm{mmse}(\gamma) = \mathbb{E} \left[ \|\mathbf{X} - \mathbb{E}[\mathbf{X}|\mathbf{Z}_\gamma]\|^2 \right]$ quantifies the minimum MSE (MMSE) at noise level $\gamma$.

Venkat and Weissman (2012) established a strong pointwise generalization of the I-MMSE identity. More recently, Kong et al. (2023) independently rediscovered the conditional form in the context of diffusion modeling:

$$\frac{d}{d\gamma} D_{\mathrm{KL}} \left( p(\mathbf{Z}_\gamma|\mathbf{X}_0) \, \| \, p(\mathbf{Z}_\gamma) \right) = \frac{1}{2} \, \mathrm{mmse}(\mathbf{X}_0, \gamma),$$

where $\mathrm{mmse}(\mathbf{X}_0, \gamma) = \mathbb{E} \left[ \|\mathbf{X} - \mathbb{E}[\mathbf{X}|\mathbf{Z}_\gamma]\|^2 \mid \mathbf{X} = \mathbf{X}_0 \right]$ is the *pointwise* MMSE.

Building on this, Kong et al. (2023) characterized the data log-likelihood in terms of denoising error:

$$-\log p_0(\mathbf{X}_0) = \frac{1}{2} \int_0^\infty \mathrm{mmse}(\mathbf{X}_0, \gamma) \, d\gamma + \mathrm{const.} \tag{7}$$

This result offers a practical and interpretable approach to likelihood estimation using the denoiser $\hat{\mathbf{X}}_\theta$ learned by the diffusion model.

Subsequently, Kong et al. (2024) extended this formulation to the conditional setting. Given auxiliary information $Y = Y_0$, the conditional negative log-likelihood admits a similar decomposition:

$$-\log p_0(\mathbf{X}_0|Y_0) = \frac{1}{2} \int_0^\infty \mathrm{mmse}(\mathbf{X}_0|Y_0, \gamma) \, d\gamma + \mathrm{const.}, \tag{8}$$

where $\mathrm{mmse}(\mathbf{X}_0|Y_0, \gamma) = \mathbb{E} \left[ \|\mathbf{X} - \mathbb{E}[\mathbf{X}|\mathbf{Z}_\gamma, Y = Y_0]\|^2 \mid \mathbf{X} = \mathbf{X}_0 \right]$ is the conditional pointwise MMSE. This enables the conditional likelihood estimation using a denoiser trained with data conditioned on the variable $Y = Y_0$, making it useful in applications such as prompt-to-output modeling in text-to-image generation.

# 3 Information-Theoretic Discrete Diffusion

The information-theoretic foundations of continuous diffusion models, reviewed in Section 2.3, motivate our exploration of analogous principles for *discrete* diffusion models. In this section, we establish a framework that links mutual information with score-based training objectives in the discrete domain. By leveraging discrete counterparts of the I-MMSE identity, we derive decompositions for data log-likelihoods. All theoretical proofs are provided in Appendix C.

## 3.1 I-MDSE Relation: An Information-Theoretic Identity for Discrete Diffusion

We now establish a discrete analog of the I-MMSE identity. In contrast to the Gaussian setting, where estimation is framed in terms of squared error, discrete diffusion models governed by the CTMC rely on score ratio estimation, captured by the denoising score entropy (DSE) loss. We show that the rate of information decay in the CTMC is governed by the *minimum* value of the DSE loss, yielding what we refer to as the **Information–Minimum Denoising Score Entropy (I-MDSE)** relation.

**I-MDSE Relation.** For the optimal score function $s_t^\star$ that minimizes the DSE loss and recovers the marginal ratio $\frac{p_t(y)}{p_t(x)}$ (Eq. (4)), we define the corresponding minimum loss as:

$$\mathrm{mdse}(t) := \min_{s_t} \mathbb{E}_{p(x_0, x_t)} \left[ \ell_{\mathrm{DSE}}(x_0, x_t, t, s_t) \right] = \mathbb{E}_{p(x_0, x_t)} \left[ \ell_{\mathrm{DSE}}(x_0, x_t, t, s_t^\star) \right],$$

where $\ell_{\mathrm{DSE}}$ is the pointwise DSE loss defined by Eq. (3).

To capture the information decay from a specific input $x_0$, we also define the *pointwise* MDSE:

$$\mathrm{mdse}(x_0, t) := \mathbb{E}_{p_{t|0}(x_t|x_0)} \left[ \ell_{\mathrm{DSE}}(x_0, x_t, t, s_t^\star) \right], \quad \text{so that} \quad \mathrm{mdse}(t) = \mathbb{E}_{p_0(x_0)} \left[ \mathrm{mdse}(x_0, t) \right].$$

We are now ready to state the discrete counterpart of the I-MMSE identity.

**Theorem 3.1** (**Pointwise and Marginal I-MDSE Relations**). *For a discrete diffusion model governed by a continuous-time Markov chain (Eq. (1)), the following pointwise I-MDSE relation holds:*

$$\frac{d}{dt} D_{\mathrm{KL}} \left( p_{t|0}(\cdot|x_0) \,\|\, p_t \right) = -\mathrm{mdse}(x_0, t). \tag{9}$$

*Taking the expectation of both sides with respect to $x_0 \sim p_0$ yields the marginal I-MDSE form:*

$$\frac{d}{dt} I(x_0; x_t) = -\mathrm{mdse}(t). \tag{10}$$

The negative sign reflects the nature of the diffusion process, in which information decays over time, since the DSE loss is always nonnegative (Lou et al., 2024). This aligns with the I-MMSE relation, where increasing SNR $\gamma$ (the inverse of time $t$) corresponds to an information gain.

**NLL Decomposition.** The I-MDSE relation further implies a decomposition of the negative log-likelihood (NLL) along the MDSE trajectory, directly paralleling Eq. (7) in the Gaussian case:

**Theorem 3.2** (NLL Decomposition via I-MDSE). *For any finite time $T > 0$, we have*

$$-\log p_0(x_0) = \int_0^T \mathrm{mdse}(x_0, t) \, dt + D_{\mathrm{KL}} \left( p_{T|0}(\cdot|x_0) \| p_T \right). \tag{11}$$

*Taking the limit $T \to \infty$ and assuming $p_T \to \pi$ for any initial $p_0$, we obtain:*

$$-\log p_0(x_0) = \int_0^\infty \mathrm{mdse}(x_0, t) \, dt. \tag{12}$$

This result reveals that the time trajectory of the DSE loss fully captures the log-likelihood of a data point. Practically, the integral in Eq. (12) can be estimated using a learned score network $s_t^\theta$ in place of the true ratio $s_t^\star$, yielding:

$$-\log p_0(x_0) \approx \int_0^\infty \mathbb{E}_{p_{t|0}(x_t|x_0)} \left[ \ell_{\mathrm{DSE}}(x_0, x_t, t, s_t^\theta) \right] dt = \lim_{T \to \infty} \mathcal{L}_{\mathrm{DSE}}^T(x_0).$$

The I-MDSE identity reveals that the commonly used DSE loss, previously viewed as a variational upper bound, is in fact an exact and theoretically grounded estimator of the log-likelihood. This equality shows that first-order score functions suffice and that no higher-order corrections are needed for likelihood estimation. Much like the I-MMSE justifies MSE in Gaussian diffusion, I-MDSE positions the DSE loss as a principled and information-theoretically sound objective in discrete diffusion, with direct implications for both training and likelihood estimation.

## 3.2 I-MDCE Relation: An Information-Theoretic Identity for Masked Diffusion

We now extend the information-theoretic analysis to masked diffusion models, where noise is applied via an absorbing process and estimation is performed through conditional prediction. In this setting, the loss of interest is the *denoising cross-entropy (DCE)* loss, which replaces score estimation with masked token reconstruction. Leveraging its pointwise equivalence to the DSE loss, we derive the **Information–Minimum Denoising Cross-Entropy (I-MDCE)** relation, the analog of the I-MMSE identity in masked diffusion. This result leads to a decomposition of the negative log-likelihood (NLL) analogous to that obtained via the I-MDSE relation.

**From DSE to DCE.** Before deriving the information-theoretic results, we first establish the pointwise equivalence between the DCE and DSE losses, which forms the basis for extending our analysis in Section 3.1 to masked diffusion.

Let $c : \{1, \ldots, N, [\mathbf{M}]\}^L \to \mathbb{R}^{L \times N}$ be a function predicting conditional distributions. We define the pointwise DCE loss as

$$\ell_{\mathrm{DCE}}(\mathbf{x}_0, \mathbf{x}, c) := \sum_{i=1}^{L} \mathbb{1}[x^i = [\mathbf{M}]] \log \frac{1}{c(\mathbf{x})_{i,x_0^i}}.$$

This loss serves as the discrete analog to squared error in the MMSE setting, measuring cross entropy (predictive accuracy) over masked positions.

We define the time-integrated DCE loss over the noise level $\Lambda \in [0, 1]$ as:

$$\mathcal{L}_{\mathrm{DCE}}^{\Lambda}(\mathbf{x}_0) := \int_0^{\Lambda} \frac{1}{\lambda} \mathbb{E}_{p_{\lambda|0}(\mathbf{x}_\lambda|\mathbf{x}_0)} \left[ \ell_{\mathrm{DCE}}(\mathbf{x}_0, \mathbf{x}_\lambda, c^\theta) \right] \, d\lambda.$$

When the conditional predictor $c$ and the score predictor $s$ are linked via the time-free reparameterization (Eq. (5)), we have:

$$s(\mathbf{x}, t)_{i,\hat{x}^i} = \frac{1 - \lambda}{\lambda} c(\mathbf{x})_{i,\hat{x}^i} \quad \text{for } \hat{x}^i \neq x^i = [\mathbf{M}],$$

where $\lambda = 1 - e^{-\overline{\sigma}(t)}$. This leads to the following equivalence of the pointwise loss functions:

**Lemma 3.3.** *If $s$ and $c$ are corresponding under time reparameterization, then*

$$\ell_{\mathrm{DSE}}(\mathbf{x}_0, \mathbf{x}, t, s_t) = \frac{\overline{\sigma}(t)(1 - \lambda)}{\lambda} \ell_{\mathrm{DCE}}(\mathbf{x}_0, \mathbf{x}, c).$$

This result establishes an exact correspondence between the time-integrated DSE and DCE losses, extending prior work that showed only asymptotic equivalence in the full-noise limit (Eq. (6)):

**Theorem 3.4** (Training Loss Equivalence)**.** *Let $\Lambda = 1 - e^{-\overline{\sigma}(T)}$ and if $s^\theta$ and $c^\theta$ are corresponding under time reparameterization. Then,*

$$\mathcal{L}_{\mathrm{DSE}}^{T}(\mathbf{x}_0) = \mathcal{L}_{\mathrm{DCE}}^{\Lambda}(\mathbf{x}_0).$$

**From I-MDSE to I-MDCE.** Having established the equivalence between the DSE and DCE losses, we now extend the information-theoretic analysis to the masked (absorbing) diffusion process. As in the DSE setting, the DCE loss is minimized by the true conditional distribution of the data:

**Theorem 3.5** (DCE Optimality)**.** *Let $c^\star$ be the data-induced conditional predictor, defined by $c^\star(\mathbf{x})_{i,\hat{x}^i} = p_0(\hat{x}^i|\mathbf{x}^{\mathrm{UM}})$. Then,*

$$c^\star = \arg\min_c \mathbb{E}_{p(\mathbf{x}_0, \mathbf{x}_\lambda)} \left[ \ell_{\mathrm{DCE}}(\mathbf{x}_0, \mathbf{x}_\lambda, c) \right].$$

Using the optimal $c^\star$, we define the *minimum DCE* (MDCE) loss and its pointwise version as:

$$\text{mdce}(\lambda) := \mathbb{E}_{p(\mathbf{x}_0, \mathbf{x}_\lambda)} \left[ \ell_{\text{DCE}}(\mathbf{x}_0, \mathbf{x}_\lambda, c^\star) \right],$$

$$\text{mdce}(\mathbf{x}_0, \lambda) := \mathbb{E}_{p_{\lambda|0}(\mathbf{x}_\lambda|\mathbf{x}_0)} \left[ \ell_{\text{DCE}}(\mathbf{x}_0, \mathbf{x}_\lambda, c^\star) \right],$$

so that $\text{mdce}(\lambda) = \mathbb{E}_{p_0(\mathbf{x}_0)}[\text{mdce}(\mathbf{x}_0, \lambda)]$.

We are now ready to state the I-MDCE relation, the masked diffusion variant of the I-MDSE identity:

**Corollary 3.6** (**Pointwise and Marginal I-MDCE Relations**). *For the absorbing diffusion model, the following identities hold:*

$$\frac{d}{d\lambda} D_{\text{KL}} \left( p_{\lambda|0}(\cdot|\mathbf{x}_0) \| p_\lambda \right) = -\frac{1}{\lambda} \text{mdce}(\mathbf{x}_0, \lambda),$$

$$\frac{d}{d\lambda} I(\mathbf{x}_0; \mathbf{x}_\lambda) = -\frac{1}{\lambda} \text{mdce}(\lambda).$$

Integrating these differential identities yields a decomposition of the log-likelihood:

**Corollary 3.7** (NLL Decomposition via I-MDCE). *For any $\Lambda \in [0, 1]$,*

$$-\log p_0(\mathbf{x}_0) = \int_0^\Lambda \frac{1}{\lambda} \text{mdce}(\mathbf{x}_0, \lambda) \, d\lambda + D_{\text{KL}} \left( p_{\Lambda|0}(\cdot|\mathbf{x}_0) \| p_\Lambda \right),$$

*and in the full-noise limit $\Lambda = 1$, this reduces to*

$$-\log p_0(\mathbf{x}_0) = \int_0^1 \frac{1}{\lambda} \text{mdce}(\mathbf{x}_0, \lambda) \, d\lambda. \tag{13}$$

In practice, replacing $c^\star$ with a learned predictor $c^\theta$ gives the estimator

$$-\log p_0(x_0) \approx \int_0^1 \frac{1}{\lambda} \mathbb{E}_{p_{\lambda|0}(\mathbf{x}_\lambda|\mathbf{x}_0)} \left[ \ell_{\text{DCE}}(\mathbf{x}_0, \mathbf{x}_\lambda, c^\theta) \right] d\lambda = \mathcal{L}_{\text{DCE}}(\mathbf{x}_0). \tag{14}$$

Similar to I-MDSE, the I-MDCE identity shows that the DCE loss used in masked diffusion training corresponds exactly to the log-likelihood, rather than serving merely as a variational upper bound. This establishes that first-order conditional predictors are sufficient for likelihood estimation in masked settings. Beyond its theoretical value as a principled foundation for training objectives, I-MDCE also enables accurate and stable likelihood estimation in practical language modeling tasks, as demonstrated in the followings sections.

## 4 Extending I-MDCE: Variants, Generalizations, and Applications

### 4.1 Time-Free Likelihood Estimation: A Variant (Alternative Formulation)

While the integral formulation of the NLL via I-MDCE (Eq. (13)) provides a solid theoretical foundation, it requires continuous integration over the diffusion coordinate. Here, we present an equivalent but more practical formulation by removing explicit time integration. This yields a time-free expression for the NLL based solely on randomly selected masked positions. [2]

**Theorem 4.1** (Time-Free Likelihood via I-MDCE). *Let $B(\cdot, \cdot)$ denote the Beta function and $H_L$ denote the $L$-th harmonic number. Then,*

$$-\log p_0(\mathbf{x}_0) = H_L \mathbb{E}_{p(I)} \left[ \sum_{i \notin I} \log \frac{1}{p_0(x_0^i | \mathbf{x}_0^I)} \right],$$

*where $\mathbf{x}_0^I$ denotes the subsequence of $\mathbf{x}_0$ consisting of the tokens indexed by $I$, and $I \subsetneq \{1, \ldots, L\}$ is the set of unmasked indices sampled from $p(I) = \frac{B(L-|I|, |I|+1)}{H_L}$.*

---

[2]Similar time-free formulations were introduced in Ou et al. (2025); Nie et al. (2025). See Appendix A.

To compute this expression in practice, we approximate the conditional distributions using the learned predictor $c^\theta$. Given a clean sequence $\mathbf{x}_0$ and a set of unmasked indices $I$, let $\tilde{\mathbf{x}}_0^I$ denote the sequence obtained from $\mathbf{x}_0$ by masking all tokens whose indices are not in $I$. We then approximate the conditional probability as

$$c^\theta(\tilde{\mathbf{x}}_0^I)_{i,x_0^i} \approx p_0(x_0^i|\mathbf{x}_0^I).$$

Using this approximation, we obtain the following time-free estimator for the likelihood:

$$-\log p_0(\mathbf{x}_0) \approx H_L \mathbb{E}_{p(I)}\left[\sum_{i \notin I} \log \frac{1}{c^\theta(\tilde{\mathbf{x}}_0^I)_{i,x_0^i}}\right]. \tag{15}$$

This formulation exhibits substantially lower variance than the time-integral form (Eq. (13)), as we empirically demonstrate in Section 5.2.

## 4.2 Conditional Likelihood Estimation: A Generalization (Structured Prediction)

The I-MDCE framework naturally extends to conditional likelihood estimation, serving as a discrete analog of Eq. (8) in the Gaussian setting, and enabling the selection of target and context components within a sequence. This is particularly useful in structured tasks such as prompt–response modeling, where the goal is to compute $\log p_0(\mathbf{x}^{I_1}|\mathbf{x}^{I_2})$ for disjoint index sets $I_1, I_2 \subseteq \{1, \ldots, L\}$.

**Theorem 4.2** (Conditional Likelihood via I-MDCE). *Let $I_1$ and $I_2$ be disjoint index sets, then*

$$-\log p_0(\mathbf{x}_0^{I_1}|\mathbf{x}_0^{I_2}) = \int_0^1 \frac{1}{\lambda} \mathbb{E}_{p_{\lambda|0}(\mathbf{x}_\lambda^{I_1}|\mathbf{x}_0^{I_1})}\left[\sum_{i \in I_1} \mathbb{1}[x_\lambda^i = [\mathbf{M}]] \log \frac{1}{p_0(x_0^i|(\mathbf{x}_\lambda^{I_1})^{\mathrm{UM}}, \mathbf{x}_0^{I_2})}\right] d\lambda. \tag{16}$$

A common example is the prompt–response setting, where the first $d$ tokens are treated as context and the remainder as the target. In practice, the integrand can be approximated using the learned conditional predictor $c^\theta$ as in the unconditional case described in Eq. (14).

Moreover, this integral form also admits a time-free equivalent based on randomly unmasking subsets of the target positions:

**Corollary 4.3** (Time-Free Conditional Likelihood via I-MDCE). *Let $I_1$ and $I_2$ be disjoint index sets and let $J$ be the randomly selected unmasked index set in $I_1$, then*

$$-\log p_0(\mathbf{x}_0^{I_1}|\mathbf{x}_0^{I_2}) = H_{|I_1|} \mathbb{E}_{p(J)}\left[\sum_{i \in I_1 \setminus J} \log \frac{1}{p_0(x_0^i|\mathbf{x}_0^{J \cup I_2})}\right], \tag{17}$$

*where the sampling distribution is $p(J) = \frac{B(|I_1|-|J|,|J|+1)}{H_{|I_1|}}$.*

In practical settings, the conditional terms $p_0(x_0^i|\mathbf{x}_0^{J \cup I_2})$ can be approximated using the trained model $c^\theta$, yielding a time-free estimator for structured conditional likelihoods analogous to Eq. (15).

## 4.3 Likelihood Ratio Estimation: An Application (Downstream Task)

Our equality-based formulation provides a principled foundation for likelihood ratio estimation using learned scores. Unlike variational bounds, which offer no guarantee when subtracted, our exact decomposition ensures that likelihood ratios can be estimated consistently and robustly. This perspective helps explain the empirical stability of recent alignment methods based on likelihood ratios in masked diffusion language models (Zhu et al., 2025).

Moreover, our time-free estimator admits a *coupled Monte Carlo* form, where a shared mask $I$ is used for both sequences:

$$\log \frac{p_0(\mathbf{y})}{p_0(\mathbf{x})} = H_L \, \mathbb{E}_{p(I)}\left[\sum_{i \notin I} \log \frac{p_0(y^i|\mathbf{y}^I)}{p_0(x^i|\mathbf{x}^I)}\right]. \tag{18}$$

Coupling via shared randomness not only ensures unbiasedness but also substantially reduces variance compared to standard decoupled estimation.

# 5 Experiments

We empirically validate the proposed I-MDCE framework through both controlled and real-world experiments. We first confirm that the time-free estimators accurately recover ground-truth likelihoods in toy settings. We then demonstrate the variance reduction effect of our estimators, showing that the time-free and coupled ratio estimators yield substantially lower Monte Carlo variance than their respective baselines. Finally, we showcase the utility of our framework in real-world tasks, including out-of-distribution detection and model influence analysis using the open-source LLaDA model (Nie et al., 2025). Further details are provided in Appendix D.

## 5.1 Reliability of Likelihood Estimation on Toy Data

This section verifies that the time-free estimators, both unconditional (Eq. (15)) and conditional (Eq. (17)), accurately recover true likelihoods in controlled toy settings.

**Unconditional Likelihood.** We first consider an unconditional setup using synthetic DNA sequences over the alphabet $\{A, T, G, C\}$. A ground-truth distribution is defined by assigning random probabilities to 128 sequences of length 8, from which one million samples are drawn to train a RADD (Ou et al., 2025) model. Figure 1a compares the true likelihoods with those estimated by Eq. (15) via Monte Carlo (MC) sampling, showing strong agreement and validating the accuracy of the unconditional estimator.

**Conditional Likelihood.** We next evaluate the conditional estimator in a more structured scenario. A long DNA sequence of length five million is generated by a 4th-order Markov chain, defining a probability distribution over all contiguous subsequences. Subsequences of length 32 are randomly sampled for training, while a held-out sequence is split into a prompt ($\mathbf{x}^{\mathrm{prompt}}$, first 16 bases) and a response ($\mathbf{x}^{\mathrm{response}}$, remaining 16 bases). We estimate $p_0(\mathbf{x}^{\mathrm{response}} | \mathbf{x}^{\mathrm{prompt}})$ via Eq. (17) and compare it to the ground-truth conditional probability from the Markov process. Figure 1b demonstrates that estimated values closely match the true likelihoods, confirming the reliability of our likelihood estimator even under complex conditional dependencies.

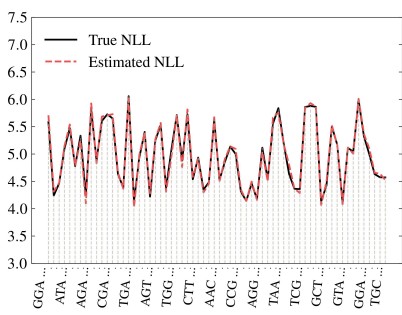

(a) Unconditional NLL via Eq. (15).

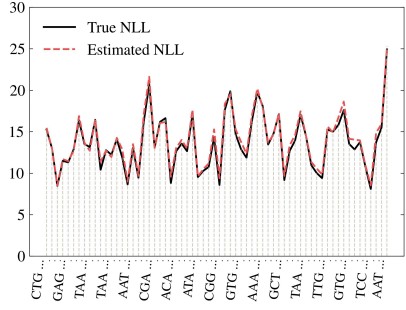

(b) Conditional NLL via Eq. (17).

Figure 1: Comparison of true and estimated NLLs on 64 sequences using our time-free estimators. Full results are provided in Appendix D.2.

## 5.2 Variance Reduction in Likelihood and Ratio Estimation

We evaluate the variance-reduction benefits of our estimators on LLaDA (Nie et al., 2025), focusing on the time-free likelihood estimator (Eq. (17)) and the coupled likelihood ratio estimator (Eq. (18)).

**Time-Free Likelihood Estimator.** We compare the variance of our time-free estimator against the time-integral baseline (Eq. (16)) by measuring the Monte Carlo variance of conditional log-likelihood estimates. As shown in Table 1a, the time-free estimator consistently achieves substantially lower variance across datasets, HellaSwag (Zellers et al., 2019), ARC-hard (Clark et al., 2018), and PIQA (Bisk et al., 2020), and for various numbers of Monte Carlo samples. These results demonstrate improved robustness and sample efficiency.

**Coupled Likelihood Ratio Estimator.** We also validate the variance-reduction effect of our coupled likelihood ratio estimator by comparing it with a standard decoupled baseline. Experiments were conducted on the BeaverTails dataset (Ji et al., 2023) using 500 prompt–response triplets ($\mathbf{x}^{\mathrm{prompt}}, \mathbf{x}^{\mathrm{response},+}, \mathbf{x}^{\mathrm{response},-}$), where $\mathbf{x}^{\mathrm{response},+}$ and $\mathbf{x}^{\mathrm{response},-}$ denote preferred and dispreferred responses, respectively. For each triplet, we estimate the log-likelihood ratio eight times to measure

Table 1: Monte Carlo variance comparison of likelihood estimators. (a) Conditional log-likelihood estimation on three datasets, with variance measured over 15 independent samples. (b) Log-likelihood ratio estimation on the BeaverTails dataset, with notably lower variance from the coupled estimator.

<div>

(a) Conditional likelihood estimation

| # MC samples | HellaSwag | | ARC_hard | | PIQA | |
|---|---|---|---|---|---|---|
| | Time-int. | Time-free | Time-int. | Time-free | Time-int. | Time-free |
| 128 | 70.97 | 11.57 | 23.18 | 5.73 | 19.77 | 4.93 |
| 256 | 30.19 | 6.02 | 18.14 | 2.96 | 15.15 | 1.81 |
| 512 | 13.38 | 2.92 | 9.50 | 1.82 | 6.50 | 1.22 |

(b) Likelihood ratio estimation

| # MC samples | Coupled | Decoupled |
|---|---|---|
| 5 | 8897.08 | 62469.41 |
| 10 | 4487.38 | 29107.21 |
| 15 | 3059.97 | 20695.61 |
| 20 | 2335.12 | 16514.72 |

</div>

empirical variance. As shown in Table 1b, the coupled estimator consistently achieves lower variance across all sample sizes, confirming its superior stability and sample efficiency.

## 5.3 Auditing and Interpretability via Conditional Likelihood Estimation

We explore the utility of our time-free conditional estimator in real-world auditing tasks aimed at inferring distributional properties of pre-trained models, such as detecting out-of-distribution (OOD) inputs or identifying training influences. These experiments show that conditional likelihood estimation provides an effective tool for interpreting model behavior.

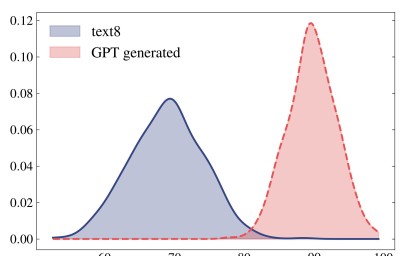

Figure 2: Estimated NLL for in-distribution (blue) and out-of-distribution (magenta). See Appendix D.4 for details.

**Detecting Out-of-Distribution Inputs.** We first test whether conditional likelihoods estimated by Eq. (17) can separate in-distribution sequences from semantically unrelated continuations. RADD (Ou et al., 2025) is trained on the `text8` corpus (Mahoney, 2011), and we compute the conditional NLL $-\log p_0(\mathbf{x}^{\text{response}}|\mathbf{x}^{\text{prompt}})$ for two response types: (1) original continuations from `text8` and (2) unrelated responses generated by GPT-4 (Achiam et al., 2023). As shown in Fig. 2, the NLL histogram reveals a clear separation: GPT-generated responses have much higher NLLs, while original continuations receive higher likelihoods. This confirms that our estimator can reliably detect OOD samples.

**Application to a Large Open-Source Model.** We further analyze input distributions using the open-source LLaDA model (Nie et al., 2025) on two datasets: `WikiText` (English) and `pretrain_zh` (Chinese). For each prompt, we estimate conditional NLLs for the original dataset continuation and for completions produced by LLaMA 3.1 (Grattafiori et al., 2024). Figure 3 shows that LLaMA 3.1-generated responses tend to receive higher average likelihoods than those from both datasets, suggesting that LLaDA may have been partially influenced by LLaMA 3.1 during training.

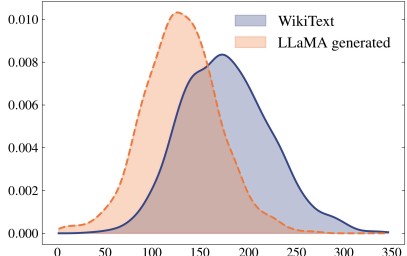

Figure 3: Estimated conditional NLL on `WikiText` (blue) and LLaMA 3.1 generated text (peach). Precise settings are in Appendix D.5.

Overall, these results highlight the utility of conditional likelihood estimation for model auditing, with natural extensions to downstream tasks such as membership inference.

## 6 Conclusion

We introduced an information-theoretic framework for discrete diffusion, formalized through the I-MDSE and I-MDCE relations that connect information decay to score-based training losses and yield exact log-likelihood decompositions. This framework offers a principled justification for learning with DSE or DCE objectives and enables practical low-variance likelihood estimation through time-free formulation. We hope this work advances the understanding of the theoretical foundations of discrete diffusion and inspires further exploration of principled estimators in generative modeling.

## Acknowledgments

This work was supported in part by Institute of Information & communications Technology Planning & Evaluation (IITP) grant funded by the Korea government (MSIT) (No. RS-2024-00457882, AI Research Hub Project), IITP grant funded by the Korean Government (MSIT) (No. RS-2020-II201361, Artificial Intelligence Graduate School Program (Yonsei University)), and the National Research Foundation of Korea (NRF) grant funded by the Korea government (MSIT) (No. RS-2025-23525649).

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

## A  Related Works

**Time-Free Likelihood Estimators.**   Time-free estimators similar to ours (Eqs. (15) and (17)) have appeared in prior work, including Ou et al. (2025) (Eq. (C.20)) and Nie et al. (2025) (Eqs. (6) and (14)). These works reformulated the variational bound $\mathcal{L}_{\text{DCE}}(\mathbf{x}_0)$ as an expectation over the number of masked tokens. In contrast, our derivation establishes this identity as an exact equality, not just a bound, and provides a quantitative comparison showing reduced variance relative to time-integral estimators.

Ou et al. (2025) further showed that the DCE loss matches the training objective of any-order autoregressive (AO-AR) models (Uria et al., 2014; Hoogeboom et al., 2022; Shih et al., 2022):

$$\mathcal{L}_{\text{AO}}(\mathbf{x}_0) = \mathbb{E}_\pi \left[ \sum_{i=1}^{L} \log \frac{1}{p_0^\theta(x_0^{\pi(i)} | \mathbf{x}_0^{\pi(<i)})} \right],$$

where the expectation is taken uniformly over all permutations of $\{1, \ldots, L\}$. While this equivalence helps explain the bidirectional behavior of masked diffusion models (Nie et al., 2025), it is computationally expensive for likelihood estimation, requiring $L$ forward passes per MC sample. In contrast, our estimator achieves the same theoretical objective using just one forward pass per sample, providing a significantly more efficient solution.

## B  Discussion and Limitations

**Conceptual Intuition.**   Although DSE/DCE and MSE originate from distinct geometries, logarithmic versus Euclidean, their connection emerges through the principle of distribution–loss matching in information theory. Just as Gaussian distributions align naturally with $\ell^2$ (MSE) loss and Laplacian distributions with $\ell^1$ (MAE) loss, categorical distributions align with logarithmic loss, which underlies DCE and, by extension, DSE in the masked diffusion setting. From this perspective, DSE and DCE serve as the natural discrete analogs of MSE, minimizing the expected divergence between predicted and true categorical distributions. This explains why the I-MDSE and I-MDCE identities carry over the information-theoretic validity of their continuous-domain counterparts.

**Limitations.**   Our framework currently applies only to masked diffusion models through the I-MDCE relation, leaving its extension to the full I-MDSE setting for future work. Moreover, while the estimator improves interpretability and auditing, its ability to recover likelihoods may also expose sensitive information, requiring cautious deployment in privacy-critical scenarios.

## C  Proofs of Theorems

### C.1  Theorem 3.1 and Theorem 3.2

This proof is strongly inspired by Lou et al. (2024)'s derivation of the variational bound for the NLL of the learned model $-\log p_0^\theta(x_0)$.

Let $\mathbb{P}$ be the path measure for the diffusion process and $\mathbb{P}_{x_0}$ be the marginalization starting from $x_0$. Using the chain rule for KL divergence of path measures (Léonard, 2014) twice (at the second and the fourth equality), we can evaluate the negative log-likelihood of the true distributions of data:

$$
\begin{aligned}
-\log p_0(x_0) &= D_{\text{KL}}(\delta_{x_0} \| p_0) \\
&= D_{\text{KL}}(\mathbb{P}_{x_0} \| \mathbb{P}) - \mathbb{E}_{x_0 \sim \delta_{x_0}}[D_{\text{KL}}(\mathbb{P}_{x_0}(\cdot|x_0) \| \mathbb{P}(\cdot|x_0))] \\
&= D_{\text{KL}}(\mathbb{P}_{x_0} \| \mathbb{P}) \\
&= D_{\text{KL}}(p_{T|0}(\cdot|x_0) \| p_T) + \mathbb{E}_{p_{T|0}(x_T|x_0)}[D_{\text{KL}}(\mathbb{P}_{x_0}(\cdot|x_T) \| \mathbb{P}(\cdot|x_T))].
\end{aligned}
$$

The last term is computed by Dynkin's formula (Hanson, 2007; Campbell et al., 2022; Lou et al., 2024), so we obtain Eq. (11):

$$
\begin{aligned}
-\log p_0(x_0) &= \int_0^T \mathbb{E}_{p_{t|0}(x_t|x_0)}[\ell_{\text{DSE}}(x_0, x_t, t, s_t^\star)]dt + D_{\text{KL}}(p_{T|0}(\cdot|x_0) \| p_T) \\
&= \int_0^T \text{mdse}(x_0, t)dt + D_{\text{KL}}(p_{T|0}(\cdot|x_0) \| p_T).
\end{aligned}
\tag{19}
$$

Letting $T \to \infty$, we obtain Eq. (12):

$$-\log p_0(x_0) = \int_0^\infty \mathrm{mdse}(x_0, t)dt.$$

Differentiating Eq. (19) with respect to $T$ and replacing $T$ with $t$, we obtain Eq. (9):

$$\frac{d}{dt}D_{\mathrm{KL}}(p_{t|0}(\cdot|x_0) \parallel p_t) = -\mathrm{mdse}(x_0, t).$$

and taking the expectation, we obtain Eq. (10):

$$\frac{d}{dt}I(x_0; x_t) = -\mathbb{E}_{p_0(x_0)}[\mathrm{mdse}(x_0, t)] = -\mathrm{mdse}(t).$$

## C.2 Lemma 3.3

$$
\begin{aligned}
\ell_{\mathrm{DSE}}(\mathbf{x}_0, \mathbf{x}, t, s_t) &= \sum_{\mathbf{y} \neq \mathbf{x}} Q_t(\mathbf{x}, \mathbf{y}) \left( s_t(\mathbf{x})_{\mathbf{y}} - \frac{p_{t|0}(\mathbf{y}|\mathbf{x}_0)}{p_{t|0}(\mathbf{x}|\mathbf{x}_0)} \log s_t(\mathbf{x})_{\mathbf{y}} + K\left(\frac{p_{t|0}(\mathbf{y}|\mathbf{x}_0)}{p_{t|0}(\mathbf{x}|\mathbf{x}_0)}\right) \right) \\
&= \sum_{x^i=[\mathbf{M}]} \sum_{y=1}^N \sigma(t) \left( \frac{1-\lambda}{\lambda} c(\mathbf{x})_{i,y} - \frac{1-\lambda}{\lambda} \mathbb{1}[y = x_0^i] \log\left(\frac{1-\lambda}{\lambda} c(\mathbf{x})_{i,y}\right) \right. \\
&\qquad\qquad\qquad\qquad \left. + \frac{1-\lambda}{\lambda} \mathbb{1}[y = x_0^i]\left(\log\left(\frac{1-\lambda}{\lambda}\mathbb{1}[y = x_0^i]\right) - 1\right) \right) \\
&= \frac{\sigma(t)(1-\lambda)}{\lambda} \sum_{x^i=[\mathbf{M}]} \left(1 - \log\left(\frac{1-\lambda}{\lambda}c(\mathbf{x})_{i,x_0^i}\right) + \log\frac{1-\lambda}{\lambda} - 1\right) \\
&= \frac{\sigma(t)(1-\lambda)}{\lambda} \sum_{x^i=[\mathbf{M}]} \log\frac{1}{c(\mathbf{x})_{i,x_0^i}} \\
&= \frac{\sigma(t)(1-\lambda)}{\lambda} \ell_{\mathrm{DCE}}(\mathbf{x}_0, \mathbf{x}, c).
\end{aligned}
$$

## C.3 Theorem 3.4

Since $\frac{d\lambda}{dt} = \sigma(t)e^{-\bar{\sigma}(t)} = \sigma(t)(1-\lambda)$, Lemma 3.3 becomes

$$\ell_{\mathrm{DSE}}(\mathbf{x}_0, \mathbf{x}, t, s_t)dt = \frac{1}{\lambda}\ell_{\mathrm{DCE}}(\mathbf{x}_0, \mathbf{x}, c)d\lambda.$$

Using the above equivalence in differential form directly, we obtain

$$
\begin{aligned}
\mathcal{L}_{\mathrm{DSE}}^T(\mathbf{x}_0) &= \int_0^T \mathbb{E}_{p_{t|0}(\mathbf{x}_t|\mathbf{x}_0)}[\ell_{\mathrm{DSE}}(\mathbf{x}_0, \mathbf{x}_t, t, s_t^\theta)]dt \\
&= \int_0^\Lambda \frac{1}{\lambda}\mathbb{E}_{p_{\lambda|0}(\mathbf{x}_\lambda|\mathbf{x}_0)}[\ell_{\mathrm{DCE}}(\mathbf{x}_0, \mathbf{x}_\lambda, c^\theta)]d\lambda \\
&= \mathcal{L}_{\mathrm{DCE}}^\Lambda(\mathbf{x}_0).
\end{aligned}
$$

## C.4 Theorem 4.1

$$-\log p_0(\mathbf{x}_0) = \int_0^1 \frac{1}{\lambda} \mathbb{E}_{p_{\lambda|0}(\mathbf{x}_\lambda|\mathbf{x}_0)} \left[ \sum_{x_\lambda^i=[\mathbf{M}]} \log \frac{1}{p_0(x_0^i|\mathbf{x}_\lambda^{\mathrm{UM}})} \right] d\lambda$$

$$= \int_0^1 \frac{1}{\lambda} \sum_{\mathbf{x}_\lambda} p_{\lambda|0}(\mathbf{x}_\lambda|\mathbf{x}_0) \sum_{x_\lambda^i=[\mathbf{M}]} \log \frac{1}{p_0(x_0^i|\mathbf{x}_\lambda^{\mathrm{UM}})} d\lambda$$

$$= \int_0^1 \frac{1}{\lambda} \sum_{I \subsetneq [L]} \lambda^{L-|I|}(1-\lambda)^{|I|} \sum_{i \notin I} \log \frac{1}{p_0(x_0^i|\mathbf{x}_0^I)} d\lambda$$

$$= \sum_{I \subsetneq [L]} \int_0^1 \lambda^{L-|I|-1}(1-\lambda)^{|I|} d\lambda \sum_{i \notin I} \log \frac{1}{p_0(x_0^i|\mathbf{x}_0^I)}$$

$$= \sum_{I \subsetneq [L]} B(L-|I|,|I|+1) \sum_{i \notin I} \log \frac{1}{p_0(x_0^i|\mathbf{x}_0^I)}.$$

To express the last formula in the expectation form, calculate the sum of the weights $B(L-|I|,|I|+1)$:

$$\sum_{I \subsetneq [L]} B(L-|I|,|I|+1) = \sum_{i=0}^{L-1} \binom{L}{i} B(L-i,i+1)$$

$$= \sum_{i=0}^{L-1} \frac{L!}{i!\,(L-i)!} \frac{(L-i-1)!\,i!}{L!}$$

$$= \sum_{i=0}^{L-1} \frac{1}{L-i}$$

$$= \sum_{j=1}^{L} \frac{1}{j}$$

$$= H_L.$$

## C.5 Theorem 4.2

In this subsection, we introduce two lemmas that directly prove Theorem 4.2.

The first lemma is quite straightforward, which is obtained by applying the diffusion process only on the indices in a nonempty subset $I$ of $\mathcal{I} = \{1, 2, \ldots, L\}$.

**Lemma C.1.** *Let $I$ be a nonempty subset of $\mathcal{I} = \{1, 2, \ldots, L\}$ and $\mathbf{x}^I = (x^i)_{i \in I}$ be the indexed subsequence of $\mathbf{x} \in \mathcal{X}^L$. Then*

$$-\log p_0(\mathbf{x}_0^I) = \int_0^1 \frac{1}{\lambda} \mathbb{E}_{p_{\lambda|0}(\mathbf{x}_\lambda^I|\mathbf{x}_0^I)} \left[ \sum_{i \in I} \mathbb{1}[x_0^i = [\mathbf{M}]] \log \frac{1}{p_0(x_0^i|(\mathbf{x}_\lambda^I)^{\mathrm{UM}})} \right] d\lambda.$$

The second lemma is obtained by regarding the data distribution with arbitrary conditioning.

**Lemma C.2.** *Under any condition $Y = y$, the negative log-likelihood is computed as*

$$-\log p(\mathbf{x}_0|y) = \int_0^1 \frac{1}{\lambda} \mathbb{E}_{p_{\lambda|0}(\mathbf{x}_\lambda|\mathbf{x}_0)} \left[ \sum_{i=1}^L \mathbb{1}[x_\lambda^i = [\mathbf{M}]] \log \frac{1}{p(x_0^i|\mathbf{x}_\lambda^{\mathrm{UM}}, y)} \right] d\lambda.$$

*Proof.* We consider the diffusion process starting from the distribution $q(\mathbf{x}_0) = p(\mathbf{x}_0|y)$ with the same noising processes $\{p_{\lambda|0}\}_{0 \le \lambda \le 1}$ as the unconditional case. Then by Eq. (13),

$$-\log q(\mathbf{x}_0) = \int_0^1 \frac{1}{\lambda} \mathbb{E}_{p_{\lambda|0}(\mathbf{x}_\lambda|\mathbf{x}_0)} \left[ \sum_{i=1}^L \mathbb{1}[x_\lambda^i = [\mathbf{M}]] \log \frac{1}{q(x_0^i|\mathbf{x}_\lambda^{\mathrm{UM}})} \right] d\lambda. \qquad (20)$$

Observe that

$$\frac{1}{q(x_0^i|\mathbf{x}_\lambda^{\text{UM}})} = \frac{q(\mathbf{x}_\lambda^{\text{UM}})}{q(x_0^i, \mathbf{x}_\lambda^{\text{UM}})} = \frac{p(\mathbf{x}_\lambda^{\text{UM}}|y)}{p(x_0^i, \mathbf{x}_\lambda^{\text{UM}}|y)} = \frac{1}{p(x_0^i|\mathbf{x}_\lambda^{\text{UM}}, y)}.$$

Substituting this result into Eq. (20) completes the proof. □

## D   Experiment Details

### D.1   Computational Resource Details

All experiments were conducted on a computing node equipped with 8 NVIDIA L40S GPUs, 1 TB of system memory, and 192 CPU cores. We used single GPU.

### D.2   Details on Toy Experiments

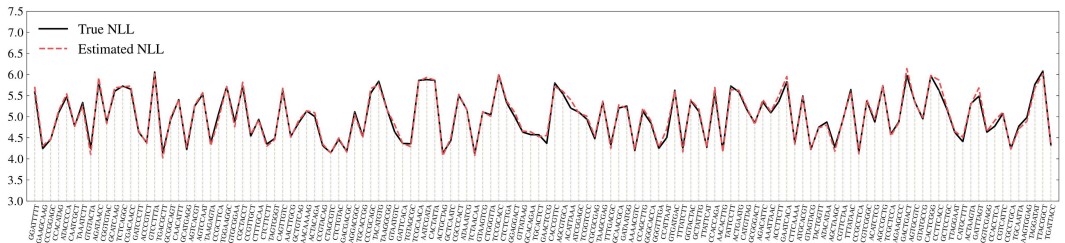

Figure 4: Results of unconditional NLL estimation on 128 DNA sequences. Estimated and true NLLs are closely aligned, supporting the effectiveness of estimation via Eq. (15).

We provide detailed experimental settings for Section 5.1, which evaluates the reliability of I-MDCE on synthetic data with an explicitly defined ground-truth distribution.

**Datasets.**   For the unconditional NLL estimation task, we generate 128 unique DNA sequences of length 8 using the alphabet $\{A, T, G, C\}$. Each sequence is assigned a probability using a softmax over uniformly sampled scores from $[0, 1)$, scaled by a temperature of 0.5. These probabilities define a categorical distribution over the 128 sequences, from which one million training samples are drawn.

For the conditional NLL estimation task, we generate sequences using a 4-th order Markov model over the same DNA alphabet. For each 4-base context, the conditional distribution over the next base is defined by the same softmax procedure applied to independently sampled scores. This results in a valid probabilistic transition table that governs sequence generation. The model is trained on a continuous DNA sequence of total length five million. For NLL evaluation, each subsequence of length 32 is split into a 16-base prompt $\mathbf{x}^{\text{prompt}}$ and a 16-base response $\mathbf{x}^{\text{response}}$.

**Training Details.**   We use the AdamW optimizer (Loshchilov and Hutter, 2019) and RADD (Ou et al., 2025) in all experiments. In the unconditional setting, the model is trained for 70,000 steps with a learning rate of $3 \times 10^{-4}$ and a batch size of 512. In the conditional setting, training is performed for 80,000 steps with a learning rate of $6 \times 10^{-4}$ and a batch size of 1,024.

**NLL Evaluation Protocol.**   When computing both conditional and unconditional NLL, we use $2^{15}$ Monte Carlo samples to estimate each case. Since this toy experiment is designed to closely align with the true data distribution, a large number of samples is used for accuracy. In general settings, however, 100 Monte Carlo samples are typically sufficient to evaluate relative differences in NLL. Full results are in Figs. 4 to 6.

### D.3   Details on Variance Reduction Experiments

We provide additional details for the variance analysis experiments described in Section 5.2.

**Conditional Likelihood Estimation.** The results in Table 1a are based on 30 randomly sampled sequences from each of the following datasets: HellaSwag (Zellers et al., 2019), ARC-hard (Clark et al., 2018), and PIQA (Bisk et al., 2020). For each sequence, we compute 15 independent Monte Carlo estimates of the conditional log-likelihood and report the variance averaged over the 30 samples. To ensure sufficient structure for conditional estimation $p_0(\mathbf{x}^{\text{response}}|\mathbf{x}^{\text{prompt}})$, we format the prompt as the question and the response as:

```
Correct: [correct answer] | Incorrect: [incorrect answer].
```

**Likelihood Ratio Estimation.** To evaluate the variance of the coupled likelihood ratio estimator in Eq. (18), we construct a dataset based on the `PKU-Alignment/BeaverTails` corpus (Ji et al., 2023). Each instance is a triplet $(\mathbf{x}^{\text{prompt}}, \mathbf{x}^{\text{response},+}, \mathbf{x}^{\text{response},-})$, where $\mathbf{x}^{\text{prompt}}$ is a prompt and $\mathbf{x}^{\text{response},+}$, $\mathbf{x}^{\text{response},-}$ are safe and unsafe responses, respectively. Specifically, we estimate the ratio of *conditional* log-likelihoods between safe and unsafe responses using the following variant of Eq. (18):

$$\log \frac{p_0(\mathbf{x}^{\text{response},+}|\mathbf{x}^{\text{prompt}})}{p_0(\mathbf{x}^{\text{response},-}|\mathbf{x}^{\text{prompt}})} = H_{|I|} \, \mathbb{E}_{p(J)} \left[ \sum_{i \in I \setminus J} \log \frac{p_0(\mathbf{x}^{\text{response},+,i}|\mathbf{x}^{\text{prompt}}, \mathbf{x}^{\text{response},J})}{p_0(\mathbf{x}^{\text{response},-,i}|\mathbf{x}^{\text{prompt}}, \mathbf{x}^{\text{response},J})} \right],$$

where $I$ denotes the index set corresponding to the response tokens and $J \subsetneq I$ is sampled from the distribution defined in Corollary 4.3. We select prompts with at least one safe and one unsafe reply, and enumerate all valid safe–unsafe response pairs per prompt to generate suitable triplets. The final dataset consists of approximately 500 triplets, formatted in JSONL with the fields:

```
{"x^prompt": prompt, "x^response,+": safe, "x^response,-": unsafe}.
```

### D.4 Training and Evaluation Details for Out-of-Distribution Detection

**Training Details.** For the OOD detection task, we train the model using a contiguous subset of the `text8` corpus (Mahoney, 2011). Each input sequence consists of 256 tokens, with the first 128 tokens serving as the conditional input $\mathbf{x}^{\text{prompt}}$ and the remaining 128 tokens as the continuation $\mathbf{x}^{\text{response}}$. We train RADD using the AdamW optimizer with a learning rate of $3 \times 10^{-4}$. The model is trained for 7,500 steps with a batch size of 32.

**NLL Evaluation Protocol.** For evaluation, we construct two groups of $(\mathbf{x}^{\text{prompt}}, \mathbf{x}^{\text{response}})$ pairs: (1) in-distribution continuations, where $\mathbf{x}^{\text{response}}$ is the true continuation of $\mathbf{x}^{\text{prompt}}$ from the held-out test split of `text8`, and (2) out-of-distribution continuations, where $\mathbf{x}^{\text{response}}$ is generated by GPT-4 given the same $\mathbf{x}^{\text{prompt}}$. All evaluation sequences are disjoint from the training data. For each group, we sample 500 examples. We then estimate NLL via Eq. (17) with 100 Monte Carlo samples, and report the distribution for each group.

### D.5 Evaluating NLL on a Pretrained Language Model

We evaluate the effectiveness of the conditional estimator (Eq. (17)) on a pre-trained open-source model. Specifically, we use the LLaDA-8B-Instruct model[3]. Two datasets are used for evaluation: `WikiText` (English) and `pretrain_zh` (Chinese). During NLL estimation, 100 MC samples are used. Results for `pretrain_zh` is in Fig. 7.

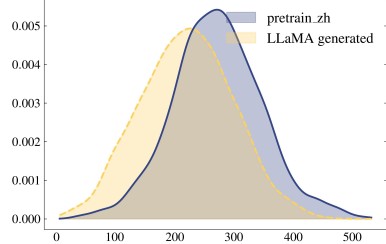

Figure 7: Estimated conditional NLL on `pretrain_zh` (blue) and LLaMA 3.1 generated text (yellow). It shows similar behavior to Fig. 3

**Evaluation Dataset.** Each dataset is preprocessed into $(\mathbf{x}^{\text{prompt}}, \mathbf{x}^{\text{response}})$ pairs, with both segments containing 64 tokens. For `WikiText`, we use the training split of `Wikitext-2-raw-v1`, while for `pretrain_zh`, we concatenate the first 3,000 documents. In both cases, the data is tokenized into 128-token blocks and evenly split into prompt and response. For each prompt $\mathbf{x}^{\text{prompt}}$, we also generate a synthetic response $\mathbf{x}^{\text{response}}$ using the LLaMA 3.1 model.

---

[3] `https://huggingface.co/GSAI-ML/LLaDA-8B-Instruct`

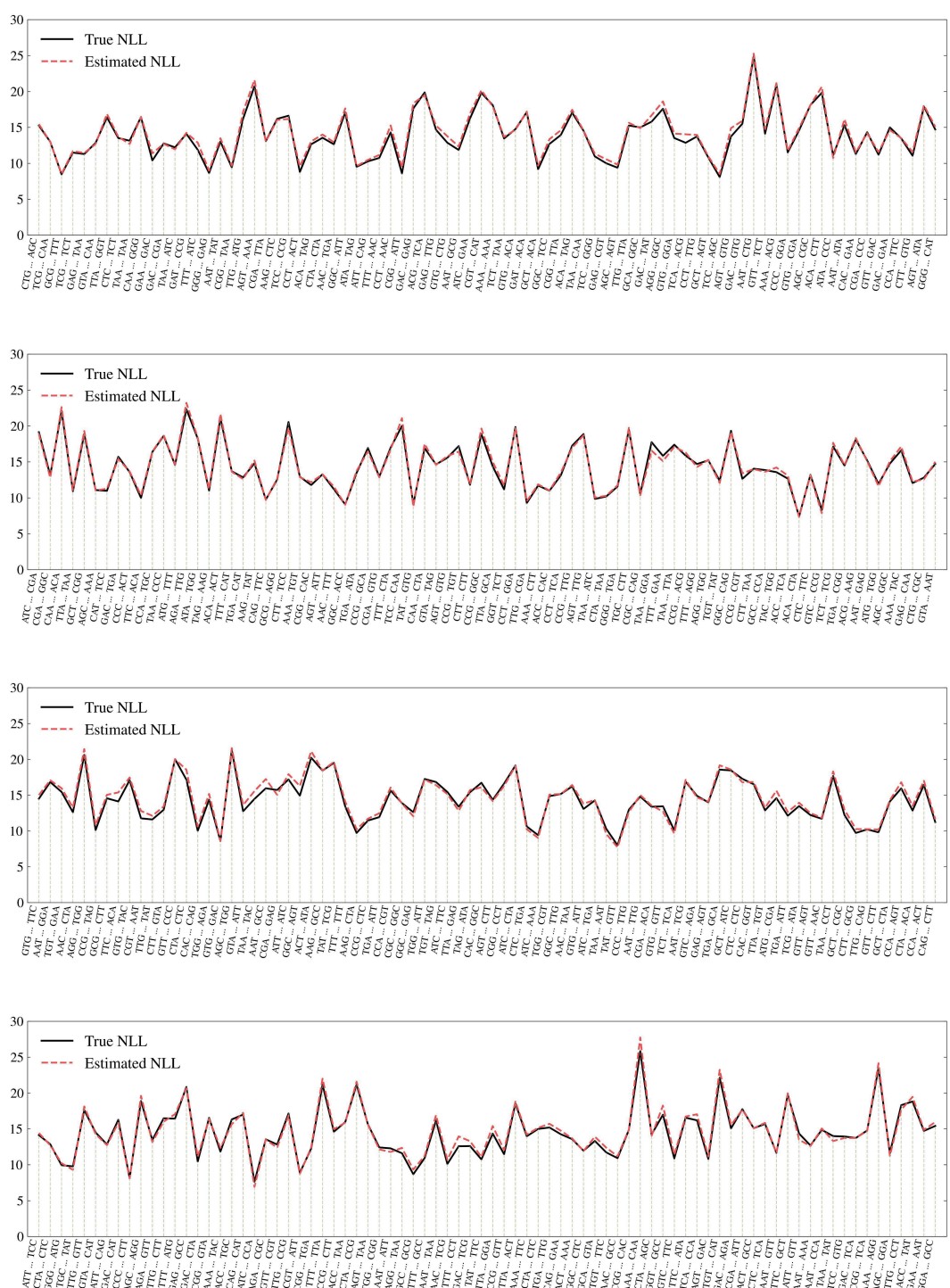

Figure 5: Conditional NLL estimation on Markov DNA sequences. Estimated and true NLLs are closely aligned, supporting the effectiveness of the estimator in Eq. (17).

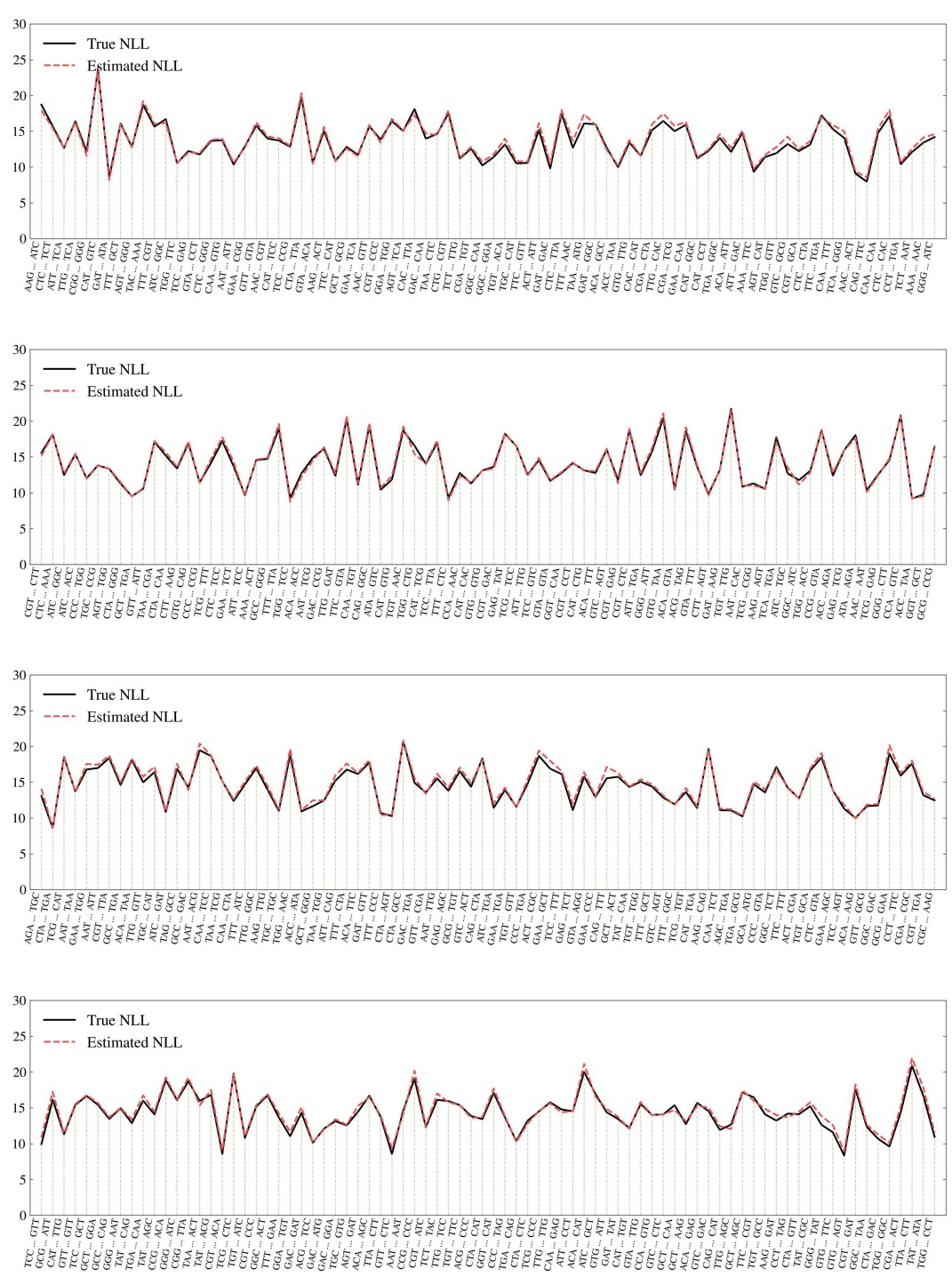

Figure 6: Conditional NLL estimation on Markov DNA sequences. Estimated and true NLLs are closely aligned, supporting the effectiveness of the estimator in Eq. (17).

