# OpenReview forum: "Information-Theoretic Discrete Diffusion"
_NeurIPS.cc/2025/Conference — NeurIPS 2025 poster_

### Official Review · Reviewer_drJV · 2025-06-26

**Clarity:** 3
**Significance:** 2
**Originality:** 3
**Rating:** 4
**Confidence:** 5

**Summary:**

This paper introduces a scalable, information-theoretic framework for discrete diffusion models, where tokens are randomly masked via an absorbing diffusion process and reconstructed using a novel info-minimization denoising cross-entropy (I-MDCE) loss that connects score-based estimation and mutual information. This approach enables efficient Monte Carlo estimation of NLL without requiring time integration. Additionally, the proposed model demonstrates strong NLL estimation on synthetic DNA data, and experiments on large models like LLaDA highlight its utility for interpreting model behavior and data influence.

**Questions:**

(Line 191) Could you clarify why the DCE can be used as an approximation to the MSE in MMSE estimation? Since DCE (cross-entropy) and MSE (L2 norm) represent fundamentally different geometric measures, what is the intuition or theoretical justification for treating them as approximately equivalent in this context?

**Ethical Concerns:**

["NO or VERY MINOR ethics concerns only"]

**Final Justification:**

Thank you for your response and for providing a variance-based analysis to support the quantitative evaluation. However, I still find that the comparison against diverse baselines is lacking, which is also raised by other reviewers. Due to this limitation, I’m inclined to lower the assessment of the paper’s significance and have adjusted my final rating to 4.

**Limitations:**

1. The time-free likelihood formulation is currently only applicable to masked diffusion models (via I-MDCE) and does not generalize to the broader I-MDSE setting.

2. The ability of the method to recover likelihoods could raise privacy concerns in sensitive applications, highlighting the need for cautious use in downstream tasks.

**Quality:**

4

**Strengths And Weaknesses:**

# Strengths:
The paper is well-structured, easy to follow, and features a thorough theoretical derivation. By integrating information theory into discrete diffusion models, it captures the essence of the information-theoretic approach, namely, building pointwise relationships, which adds substantial novelty. The authors are the first to introduce the I-MDSE relationship. To further extend this, they derive a mask setting of I-MDSE in masked diffusion models, termed I-MDCE. Additionally, to improve scalability, they propose a time-free likelihood estimation framework to facilitate model training.


# Weaknesses:
- The experimental section is relatively limited. It only validates the effectiveness of the proposed model, without comparison to other baselines. For example, in the experiments on DNA sequences, could NLL estimation be performed using other discrete diffusion models as well? Beyond the visualizations in Figures 1 and 2, is it possible to provide concrete quantitative metrics to measure the estimation error? I suggest expanding the experiments, such as displaying sampling results in the prompt-response experiments and comparing perplexity across different models. Nevertheless, given the theoretical contributions of this paper, the current experiments are still acceptable.

- A few minor points to note:

  - There is a comma typo in Eq. (10) and Eq. (12).

  - It is recommended to number all the equations on page 6.

  - In Line 214, the description is reversed. It should read: "In practice, replacing $c^*$ with a learned predictor $c^\theta$ gives the estimator."

---

> ### Author Rebuttal · Authors · 2025-07-31
>
> > **W1. The experiments are limited in scope and lack baseline comparisons or quantitative metrics.**
>
> We sincerely thank the reviewer for the thoughtful feedback and for highlighting the need for broader experimental validation. We fully agree that adding stronger baseline comparisons and quantitative metrics is essential to support the practical effectiveness of our framework. While the reviewer specifically mentioned DNA and prompt-response tasks, we instead chose to evaluate our estimator in more realistic, large-scale settings using the LLaDA model [1], in order to better reflect practical use cases beyond synthetic settings. To this end, we conducted two additional experiments focused on variance reduction and likelihood ratio estimation.
>
> 1. Variance comparison of likelihood estimators.\
> We compared our time-free estimator (Eq. 14) with the time-integral estimator (Eq. 13), which corresponds to the standard variational bound. We evaluated the average of Monte Carlo (MC) estimations of conditional log-likelihood estimates on 30 sequences with 15 samples each from HellaSwag[2], ARC_hard[3], and PIQA[4] using the LLaDA model. As shown in the following table, the time-free estimator consistently yields lower variance:
>
> ### Variance ↓ of Estimator (HellaSwag, 30 subset)
> | #MC sample | Time-integral | Time-free |
>  | ----- | ----- | ----- |
> | 128 | 70.9716 | 11.5658 |
> | 256 | 30.1938 | 6.0150 |
> | 512 | 13.3796 | 2.9170 |
> ---
> ### Variance ↓ of Estimator (ARC_hard, 30 subset)
> | #MC sample | Time-integral | Time-free |
>  | ----- | ----- | ----- |
> | 128 | 23.1758 | 5.7318 |
> | 256 | 18.1444 | 2.9612 |
> | 512 | 9.5014 | 1.8200 |
> ---
> ### Variance ↓ of Estimator (Piqa, 30 subset)
> | #MC sample | Time-integral | Time-free |
>  | ----- | ----- | ----- |
> | 128 | 19.76875 | 4.92825 |
> | 256 | 15.14975 | 1.80950 |
> | 512 | 6.50375 | 1.22425 |
>
>  This finding aligns with observations in the LLaDA paper:
>
> *"Eq. (12) (RADD formulation) requires over 1000 Monte Carlo estimates for stable results, whereas Eq. (14) (LLaDA’s reformulation) achieves stability with only 128 estimates."*
>
> 2. Likelihood ratio estimation via coupled sampling.\
> Beyond what is discussed in the paper, our time-free formulation also offers an additional practical advantage: it supports a unified (coupled) estimator for likelihood ratios, enabled by an explicit sampling distribution over masking positions $J$. This allows computing $\log p(y^+|x)$ and $\log p(y^-|x)$ under shared randomness, avoiding the high variance of naive subtraction:
> $$
> \log\frac{p(y^+|x)}{p(y^-|x)} = H\_L \mathbb{E}\_J \left[ \sum_{i\in I\setminus J}\log\frac{p(y^{+,i}|y^{+,J},x)}{p(y^{-,i}|y^{-,J},x)} \right]
> $$
> Here, $x$ denotes a prompt, and $y^+$ and $y^-$ represent the preferred and dispreferred completions, respectively. This setup commonly arises in alignment tasks where a model is trained to score preferred responses higher than dispreferred ones. In particular, the Direct Preference Optimization (DPO) objective [5] optimizes the objective that consists of the following alignment score:
> $$
> \log\frac{p^\theta(y^+|x)}{p^\theta(y^-|x)}
> $$
> which corresponds to the log-likelihood ratio between the preferred and dispreferred responses under the current model $p^\theta$. This form naturally motivates the use of likelihood ratio estimators, especially in settings where the responses $y^+$ and $y^-$ are sampled from human preference data.\
> To validate this, we conducted a likelihood ratio estimation experiment on the BeaverTails [6] dataset, which contains 500 $(x, y^+, y^-)$ triples annotated with human preferences. The average variance of MC estimates was consistently lower for our coupled estimator, as shown in the following table:
>
> ### Variance ↓ of Log-Likelihood Ratio Estimation (BeaverTails, 500 triples)
> |#MC samples | Coupled       | Decoupled      |
> |-------|--------------|--------------|
> | 5     | 8897.079131 | 62469.413827 |
> | 10     | 4487.376690 | 29107.212740 |
> | 15     | 3059.972797 | 20695.614707 |
> | 20     | 2335.123036 | 16514.715472 |
>
> These results demonstrate that our estimator not only improves sample efficiency in likelihood estimation but also enables stable and low-variance ratio estimation, both of which are critical for real-world applications involving alignment, robustness, and interpretability.
>
> > **W2. Minor points including typos.**
>
> Thank you for pointing out these issues. In addition to correcting the comma typos in Eq. (10) and Eq. (12), numbering all equations on page 6, and fixing the reversed phrasing in Line 214 as suggested, we will take this opportunity to carefully review the entire manuscript for any remaining minor errors.
>
> In particular, we sincerely apologize for having mistakenly reversed the first and last name of the lead author of the RADD paper. We will correct this and ensure that all references are accurate and respectful in the final version.
>
> We greatly appreciate your attentive reading and helpful suggestions.
>
> > **Q1. Clarify why DCE can approximate MSE in MMSE estimation despite their different geometric interpretations.**
>
> Thank you for this thoughtful question. While it's true that cross-entropy (DCE) and mean squared error (MSE) arise from fundamentally different geometric interpretations (logarithmic vs. Euclidean) their use as estimation objectives can be understood through the lens of distribution-loss matching, as commonly seen in information theory and estimation theory.
>
> For example, logarithmic loss is the natural loss function when modeling data with a discrete distribution, just as L2 loss aligns with Gaussian distributions and L1 loss with Laplacian distributions. In our setting, the cross-entropy (DCE) loss corresponds to logarithmic loss, which is the proper scoring rule for categorical (discrete) output distributions, and is directly tied to pointwise mutual information and entropy minimization. This connection has been theoretically formalized in works such as Courtade & Weissman (2013) and Jiao et al. (2015) [7, 8].
>
> Therefore, while DCE and MSE measure different quantities, DCE plays a role analogous to MSE in the discrete domain: it minimizes expected divergence between the predicted distribution and the true discrete target, just as MSE minimizes distance under a continuous Gaussian model. This analogy justifies the use of DCE as a surrogate for MMSE in discrete diffusion settings, particularly when the underlying variable is categorical.
>
> We will clarify this theoretical motivation in the revised manuscript and cite the appropriate references.
>
> [1] Nie et al., Large language diffusion models, arXiv, 2025.\
> [2] Zellers et al., HellaSwag: Can a Machine Really Finish Your Sentence?, ACL, 2019.\
> [3] Clark et al., Think you have Solved Question Answering? Try ARC, the AI2 Reasoning Challenge, arXiv, 2018.\
> [4] Bisk et al., PIQA: Reasoning about Physical Commonsense in Natural Language, AAAI, 2020.\
> [5] Rafailov et al., Direct Preference Optimization: Your Language Model is Secretly a Reward Model, NeurIPS, 2023.\
> [6] Nie et al., BeaverTails: Evaluating Language Models by Simulating Human Preference Data, arXiv, 2024.\
> [7] Thomas and Weissman, Multiterminal Source Coding under Logarithmic Loss, IEEE Trans. Inf. Theory, 2013.\
> [8] Jiao et al., Justification of Logarithmic Loss via the Benefit of Side Information, IEEE Transactions on Information Theory, 2015

---

### Official Review · Reviewer_wh8N · 2025-06-28

**Clarity:** 2
**Significance:** 3
**Originality:** 2
**Rating:** 4
**Confidence:** 3

**Summary:**

This paper proposes a framework introducing information-theoretic framework for discrete diffusion models, by using principled estimators of log-likelihood based on score-matching losses. Empirical results on synthetic and real-world data confirm that the proposed approach accurately estimates log-probabilities of discrete inputs.

**Questions:**

Could the authors provide baseline comparisons to better support the effectiveness of the proposed framework?

**Ethical Concerns:**

["NO or VERY MINOR ethics concerns only"]

**Final Justification:**

While a more detailed comparison against diverse baselines would have been helpful, I appreciate the authors' thoughtful clarifications and the additional experimental results provided during the rebuttal. In light of the strengths highlighted by other reviewers—particularly the use of discrete diffusion via I-MMSE connections—I have decided to raise my rating to align with their perspectives.

**Limitations:**

Yes

**Quality:**

3

**Strengths And Weaknesses:**

**Strength**

* The submission proposed I-MDSE and I-MDCE that connect mutual information decay with score-based loss functions in discrete diffusion.

* The author provides closed-form of negative log-likelihood time-integral and time-free NLL estimators  for general discrete diffusion and masked diffusion and extends the NLL estimation to the conditional setting for structured prediction tasks..

**Weaknesses**

* The experimental scale is limited, and the absence of comparisons with prior work (e.g. baseline comparison) makes it difficult to fully validate the strength of the proposed framework. For instance, compared to other similar approaches such as ItDPDM (Information-Theoretic Discrete Poisson Diffusion Model), which includes comprehensive real-world evaluations and baselines to support its claims, this submission lacks such empirical breadth.

* This framework appears to be an extension of Information-Theoretic Diffusion, adapting its analogous results to discrete diffusion settings. However, without clear motivation, detailed explanations, or illustrative examples, it is difficult to assess whether this submission offers more than an incremental contribution.

---

> ### Author Rebuttal · Authors · 2025-07-31
>
> > **W1. The experimental evaluation lacks scale and baseline comparisons, making it difficult to validate the practical strength of the proposed framework compared to prior work like ItDPDM.**
>
> We thank the reviewer for the thoughtful comment and for raising the important point regarding experimental scale and baseline comparisons.
>
> We acknowledge that our empirical evaluation is more limited in scope compared to broader real-world studies such as the Information-Theoretic Discrete Poisson Diffusion Model (ItDPDM), which presents a novel noise process and includes extensive benchmarking. That work explores a valuable modeling direction.
>
> Our paper, however, pursues a different objective: rather than proposing a new diffusion model, we focus on extending information-theoretic tools (in particular, I-MMSE-style decomposition identities) to the discrete and masked diffusion setting, which is widely adopted in real-world applications such as LLaDA. Our main contribution lies in the derivation of exact equality-based decompositions (I-MDSE and I-MDCE) that enable principled estimation of log-likelihood and, critically, log-likelihood ratios.
>
> While the original submission emphasized the theoretical contributions and their use in conditional likelihood estimation, we agree that additional empirical validation is essential. In this rebuttal, we have added new experiments demonstrating two key advantages of our formulation:
> - Variance reduction of the time-free estimator compared to the time-integral baseline (RADD-style).
> - A unified estimator for likelihood ratios that achieves lower variance than naïve decoupled subtraction.
>
>
> These new results directly support the practical strength of our approach and are further detailed in our response to Q1.
>
> We will revise the paper to more clearly distinguish our estimator-centric, theory-driven focus from prior work centered on model design, such as ItDPDM, and to transparently acknowledge the current experimental scope while highlighting the newly added empirical support.
>
>
> > **W2. The work adapts known information-theoretic results to the discrete diffusion setting, but lacks clear motivation, making it hard to determine whether the contribution is truly substantive or merely incremental.**
>
> We thank the reviewer for the comment and the opportunity to clarify the motivation and significance of our theoretical contribution.
>
> At the core of our work are the I-MDSE (Theorem 3.1) and I-MDCE (Corollary 3.6) identities, which we consider to be the central theoretical contributions of the paper. These identities establish exact equalities between mutual information decay and score-based losses in discrete and masked diffusion processes. This equality-based structure mirrors a fundamental theme in classical information and estimation theory, for instance, the I-MMSE identity connects Gaussian noise with MSE loss, offering deep insight into the behavior of continuous diffusion models. Our work extends this paradigm to the categorical setting, showing that cross-entropy and score entropy losses serve as the natural analogs under discrete noise.
>
> We believe these results are not only conceptually meaningful, but also practically impactful. In particular, the I-MDSE and I-MDCE identities underpins Theorem 3.2 and Corollary 3.7, which show that commonly used variational training losses are not merely loose bounds, but are in fact tight equalities in our framework. This reveals that first-order score functions are sufficient to fully capture the log-likelihood, placing a theoretical limit on the benefits of higher-order corrections. We view this as a substantial theoretical insight that reframes how variational objectives should be understood in discrete diffusion models.
>
> Moreover, this exactness directly enables practical extensions. Theorem 3.2 and Corollary 3.7 serve as the theoretical foundation for our low-variance, time-free log-likelihood estimator, introduced in the main paper, and further support our newly proposed likelihood ratio estimator, which is presented for the first time in this rebuttal. These estimators go beyond standard variational approximations. The ratio estimator, in particular, is empirically validated through new experiments (see Q1), demonstrating that the equality results not only offer theoretical elegance but also lead to robust and efficient tools for real-world tasks such as alignment, OOD detection, and membership inference.
>
> In summary, our work represents a principled generalization of foundational information-theoretic tools to the discrete and masked diffusion setting, which is increasingly relevant in modern generative modeling. We will revise the manuscript to better articulate the motivation, tightness, and practical implications of these equality-based results.
>
>
>
> > **Q1. Could the authors provide baseline comparisons to better support the effectiveness of the proposed framework?**
>
> We appreciate the reviewer’s suggestion and have included baseline comparisons to better support the effectiveness of our proposed framework.
> First, we conducted a direct comparison between our time-free estimator (Eq. 14) and the time-integral estimator (Eq. 13) to assess the practical utility of the former. Specifically, we measured the Monte Carlo (MC) variance of conditional log-likelihood estimates on 30 sequences with 15 samples each from HellaSwag[1], ARC_hard[2], and PIQA[3] using the LLaDA model. As shown in the following table:
> ### Variance ↓ of Estimator (HellaSwag, 30 subset)
> | #MC sample | Time-integral | Time-free |
>  | ----- | ----- | ----- |
> | 128 | 70.9716 | 11.5658 |
> | 256 | 30.1938 | 6.0150 |
> | 512 | 13.3796 | 2.9170 |
> ---
> ### Variance ↓ of Estimator (ARC_hard, 30 subset)
> | #MC sample | Time-integral | Time-free |
>  | ----- | ----- | ----- |
> | 128 | 23.1758 | 5.7318 |
> | 256 | 18.1444 | 2.9612 |
> | 512 | 9.5014 | 1.8200 |
> ---
> ### Variance ↓ of Estimator (Piqa, 30 subset)
> | #MC sample | Time-integral | Time-free |
>  | ----- | ----- | ----- |
> | 128 | 19.76875 | 4.92825 |
> | 256 | 15.14975 | 1.80950 |
> | 512 | 6.50375 | 1.22425 |
>
> The results shows that the time-free estimator consistently exhibits lower variance. This finding aligns with observations in the LLaDA paper:
>
> "Eq. (12) (RADD formulation) requires over 1000 Monte Carlo estimates for stable results, whereas Eq. (14) (LLaDA’s reformulation) achieves stability with only 128 estimates."
>
> Beyond variance reduction in marginal likelihood estimation, our time-free formulation also enables a key extension not addressed by prior work: unified (i.e., coupled) likelihood ratio estimation. This is possible because our formulation defines an explicit sampling distribution over mask positions $J$, allowing us to compute both log $p(y^+|x)$ and log $p(y^-|x)$ under shared randomness. Rather than computing two separate estimates and subtracting them, which often leads to compounded variance, our method yields a direct estimator of the log-likelihood ratio with provable correctness and improved stability:
>
> $$
> \log\frac{p(y^+|x)}{p(y^-|x)} = H\_L \mathbb{E}\_J \left[ \sum_{i\in I\setminus J}\log\frac{p(y^{+,i}|y^{+,J},x)}{p(y^{-,i}|y^{-,J},x)} \right]
> $$
>
> where $I$ denotes the index set of response tokens, $L = |I|$, and $J$ is sampled as described in Corollary 3.10.
>
> This property is particularly valuable in downstream tasks where likelihood ratios are central. For example, Direct Preference Optimization (DPO) [4] directly optimizes log-likelihood ratios between preferred and dispreferred responses, and likelihood ratio tests are fundamental to out-of-distribution (OOD) detection [5,6].
>
> To evaluate the empirical benefits, we conducted a likelihood ratio estimation experiment on the BeaverTails dataset [7]. Our estimator, using coupled MC sampling, achieved significantly lower variance compared to the standard decoupled (two-sample) baseline. When computing log-likelihood ratios for 500 $(x, y^+, y^-)$ triples, the average variance was consistently lower, as shown in the following table:
>
> ### Variance ↓ of Log-Likelihood Ratio Estimation (BeaverTails, 500 triples)
> |#MC samples | Coupled       | Decoupled      |
> |-------|--------------|--------------|
> | 5     | 8897.079131 | 62469.413827 |
> | 10     | 4487.376690 | 29107.212740 |
> | 15     | 3059.972797 | 20695.614707 |
> | 20     | 2335.123036 | 16514.715472 |
>
> Finally, to demonstrate applicability at scale, we apply our framework to the large-scale open-source LLaDA model (Section "Application to a Large Scale Open-Source Model"). In this setting, we use our conditional likelihood estimator for a membership inference task, comparing completions from LLaDA with real data across English and Chinese datasets. The results reveal signs of training data overlap, highlighting the practical relevance of our estimator for tasks involving privacy auditing and model interpretability.
>
> We believe these comparisons and experiments provide strong empirical justification for the advantages of our time-free formulation, both in terms of estimator quality and its applicability to important real-world tasks.
>
> [1] Zellers et al., HellaSwag: …, ACL, 2019.\
> [2] Clark et al., Think you have solved question answering?..., arXiv, 2018.\
> [3] Bisk et al., PIQA…, AAAI, 2020.\
> [4] Rafailov et al., Direct preference optimization: Your language model is secretly a reward model. NeurIPS, 2023.\
> [5] Ren et al., Likelihood ratios for out-of-distribution detection. NeurIPS, 2019.\
> [6] Zhang et al., Your Finetuned Large Language Model is Already a Powerful Out-of-distribution Detector. AISTATS, 2025.\
> [7] Nie et al., BeaverTails: Evaluating Language Models by Simulating Human Preference Data. arXiv, 2024.

---

> > ### Comment · Reviewer_wh8N · 2025-08-03
> >
> > Thank you for the detailed rebuttal and additional experiments. While the new results help clarify aspects of the contribution, my core concerns remain. The empirical evaluation is still limited in scale and the theoretical extensions, though interesting, are not clearly motivated or demonstrated to offer more than incremental value. I will therefore maintain my original score.

---

### Official Review · Reviewer_pwj1 · 2025-06-30

**Clarity:** 4
**Significance:** 2
**Originality:** 3
**Rating:** 4
**Confidence:** 5

**Summary:**

I-MMSE relations in information theory establish exact connections between density estimation and denoising in a Gaussian channel.
This paper generalizes these ideas to consider discrete random variables and discrete noise channels (like masking), demonstrating the existence of exact relations between probability and optimal denoising in these discrete channels, using the correct objective.
Denoising with discrete noise channels has become popular recently in the discrete diffusion literature. This paper establishes that variational objectives that were previously understood as bounds on probability are actually tight.
The authors use their results to express probability estimates in different forms, and apply these estimates to synthetic and real-world examples.

**Questions:**

My main question is related to the discussed weaknesses. Is there anything about your NLL estimator that makes it significantly different in implementation or performance compared to existing methods, and are there any experiments that demonstrate this?

**Ethical Concerns:**

["NO or VERY MINOR ethics concerns only"]

**Final Justification:**

Several reviewers found the experiments to be limited, because of lack of comparisons to other methods. The authors discussed comparisons and gave some more variance estimates in the rebuttal. I raised my score to borderline accept, but I do still consider this a borderline submission with the current evaluation.

**Limitations:**

yes

**Quality:**

3

**Strengths And Weaknesses:**

related):
## Strengths:

- I thought the summary of discrete diffusion and I-MMSE connections were nicely presented.
- Even though the result is functionally the same as the variational bound, I like the point that Eq. 11 shows that it "fully captures the log-likelihood".  To me, this is quite important as it shows that it is not possible/necessary to tighten the variational bound by outputting higher order moments.
- I appreciate the effort to make synthetic experiments with known ground truth that are somewhat interesting (DNA-like, higher order Markov).


## Weaknesses:

### Significance of time-free formulation

> In contrast, we derive a surprising result: a time-free expression for the NLL based solely on randomly selected masked positions. To the best of our knowledge, such a time-free formulation has not been established in any diffusion model, whether discrete or continuous.

$$ - \log p_0\left( \mathbf{x}_0^{I_1} \mid \mathbf{x}_0^{I_2} \right) = H_{|I_1|} \, \mathbb{E}_{p(J)} \left[ \sum_{i \in I_1 \setminus J} \log \frac{1}{p_0\left(x^i \mid \mathbf{x}_0^{J \cup I_2} \right)} \right] $$

This seems like an over-statement, or needs more context about what the surprising new claim is. Many papers have recognized that NLL can be written in terms of random order autoregression.

- E. Hoogeboom, A. A. Gritsenko, J. Bastings, B. Poole, R. v. d. Berg, and T. Salimans,
“Autoregressive diffusion models,” arXiv preprint arXiv:2110.02037, 2021.
- A. Pannatier, E. Courdier, and F. Fleuret, “σ-gpts: A new approach to autoregressive models,”
in Joint European Conference on Machine Learning and Knowledge Discovery in Databases,
pp. 143–159, Springer, 2024.
- Z. Yang, Z. Dai, Y. Yang, J. Carbonell, R. R. Salakhutdinov, and Q. V. Le, “Xlnet: Generalized
autoregressive pretraining for language understanding,” Advances in neural information
processing systems, vol. 32, 2019.


This paper explicitly is about how time-dependence for masked diffusion models is not needed.

- K. Zheng, Y. Chen, H. Mao, M.-Y. Liu, J. Zhu, and Q. Zhang, “Masked diffusion models are
secretly time-agnostic masked models and exploit inaccurate categorical sampling,” arXiv
preprint arXiv:2409.02908, 2024.

You also cite RADD, which says something similar in Eq. 3.8. Maybe the difference between your statement and RADD is that they sample the masks based on sampling a continuous "time variable", while you sample directly from a discrete mask distribution?
Even so, the regular any-order AR decomposition is "time-free" in the same way. Given a random permutation, $i_1, ...,i_T = \pi(1, ... T)$

$$-\log p(x_{i_T:i_k} \mid x_{i_{k-1}:i_1}) =   \sum_{j =k}^T \log \frac{1}{p\left(x_{i_j} \mid x_{i_{j-1}:i_1} \right)}$$

By averaging over random orders, you can get the same result as yours - arguably more general, as we could weight this to allow for different mask distributions.

### Experimental results missing useful comparisons

My main concern is that the experimental section is not very useful and may even be misleading.
We already have many ways to estimate the NLL, using cross entropy with random order autoregressive and masked models and with variational bounds.
What would be useful to know is whether the results in this paper help us to do this easier or better.

The reason it seems misleading is that your estimator may be giving identical results to existing estimators. But because baselines are not presented, it seems as if these tasks were enabled by your results.

Types of baselines for bounding NLL I would have expected to see:
- Random order Autoregressive (one sample, or averaged over orders)
- Variational bounds like DSE

Types of results I would have expected to see:
- Bias and variance of various estimators
- Computational cost, if relevant

The random tasks like OOD are neat demos, but I'd expect any of the existing NLL bounds would be just as good for these.

---

> ### Author Rebuttal · Authors · 2025-07-31
>
> > **W1. The novelty and practical benefit of the proposed time-free NLL formulation is unclear.**
>
> We appreciate the reviewer’s question regarding the novelty and practical benefit of our proposed time-free NLL formulation, especially in relation to RADD, LLaDA, and any-order autoregressive models.
>
> First, we clarify that our time-free formulation differs from RADD, which expresses the NLL as an expectation over a continuous time variable $\lambda$. In contrast, our estimator is derived over a discrete mask distribution over $J$, removing the need for time-based integration. This reparameterization contributes to reduced variance, as previously hinted at in follow-up works.
> That said, we acknowledge that our original claim overstated the novelty. The LLaDA paper Eq.(14) [1] introduces a formulation that is closely related to ours: it reinterprets the RADD estimator as an expectation over the number of masked tokens ($\ell$), rather than time. While LLaDA focused only on upper bounds and did not prove equality, the resulting structure is indeed similar to our time-free form. LLaDA briefly notes that this formulation empirically lowers variance:
>
> *"Eq. (12) (RADD formulation) requires over 1000 Monte Carlo estimates for stable results, whereas Eq. (14) (LLaDA’s reformulation) achieves stability with only 128 estimates."*
>
> Building on that, we provide a more rigorous experimental evaluation and demonstrate that our time-free estimator achieves significantly lower variance, as shown in the table below. In the revised version of the paper, we will explicitly clarify the connection and differences between RADD, LLaDA, and our work.
>
> We conducted a direct comparison between our time-free estimator (Eq. (14)) and the time-integral estimator (Eq. (13)) to further assess the practical utility of the former. Specifically, we measured the Monte Carlo (MC) variance of conditional log-likelihood estimates on 30 sequences with 15 samples each from HellaSwag[6], ARC_hard[7], and PIQA[8] using the LLaDA model. As shown below, the time-free estimator consistently exhibits lower variance:
>
> ### Variance ↓ of Estimator (HellaSwag, 30 subset)
> | #MC sample | Time-integral | Time-free |
>  | ----- | ----- | ----- |
> | 128 | 70.9716 | 11.5658 |
> | 256 | 30.1938 | 6.0150 |
> | 512 | 13.3796 | 2.9170 |
> ---
> ### Variance ↓ of Estimator (ARC_hard, 30 subset)
> | #MC sample | Time-integral | Time-free |
>  | ----- | ----- | ----- |
> | 128 | 23.1758 | 5.7318 |
> | 256 | 18.1444 | 2.9612 |
> | 512 | 9.5014 | 1.8200 |
> ---
> ### Variance ↓ of Estimator (Piqa, 30 subset)
> | #MC sample | Time-integral | Time-free |
>  | ----- | ----- | ----- |
> | 128 | 19.76875 | 4.92825 |
> | 256 | 15.14975 | 1.80950 |
> | 512 | 6.50375 | 1.22425 |
>
> These findings empirically validate the variance-reduction benefit of our formulation and align with observations made in LLaDA.
>
> We also agree that our formulation is conceptually related to any-order autoregressive (AR) decompositions, such as those in XLNet and $\sigma$-GPTs. This connection is noted in LLaDA, and we will explicitly clarify this relationship in the revised manuscript. However, we emphasize an important distinction: any-order AR estimation requires a full permutation of length $L$ and evaluation of conditional probabilities for each position,
>
> $$p\left( X\_{\sigma_T} = x\_{\sigma_T} \mid X_{\sigma_1} = x_{\sigma_1}, \ldots, X_{\sigma_{T-1}} = x_{\sigma_{T-1}} \right)$$
>
> resulting in $L$ forward passes for sequences of length $L$.
>
>
>  In contrast, our method supports per-sample forward computation, only one forward pass per Monte Carlo sample, and remains stable even with fewer samples than sequence length (e.g., 128 samples for a sequence of length 190).
>
> We appreciate the reviewer for pointing out these valuable connections, and we will revise the manuscript to more clearly articulate the positioning, distinctions, and practical contributions of our approach relative to these related works.
>
> > **W2. Concern about missing baseline comparisons, how the proposed estimator compares to existing methods, and whether it offers practical improvements.**
>
> We appreciate the reviewer’s concern regarding the absence of baseline comparisons. To address this, we have added experiments comparing our estimator to RADD. The results (see table below) demonstrate that our time-free estimator achieves significantly lower variance, highlighting its robustness and sample efficiency. (See W1)
>
> These results also suggest a potential limitation when applying AO-AR estimators to tasks with longer responses. For example, in our experiment with HellaSwag, where the average response length is 228 tokens, a single MC sample under AO-AR requires roughly 228 forward passes, meaning that a 256-NFE budget yields only about 1 MC sample. While AO-AR remains a principled objective, this sampling cost may make it less practical in settings where efficient estimation is needed.
>
> Importantly, our method provides an exact equality, not just an upper bound, for the log-likelihood. This distinction is crucial: because the formulation holds as an equality, it allows us to construct a direct estimator for the log-likelihood ratio, a task that is not straightforward when using variational upper bounds. Likelihood ratio estimation plays a central role in large language models (LLMs), with numerous downstream applications. For example, Direct Preference Optimization (DPO) [2, 3], a prominent alignment algorithm for LLMs, directly optimizes a loss based on the log-likelihood ratio between preferred and dispreferred outputs. More specifically, given a prompt $x$ with a preferred output $y^+$ and a dispreferred output $y^-$, the DPO loss optimizes an alignment score of the following form:
> $$
> \log\frac{p^\theta(y^+|x)}{p^\theta(y^-|x)}
> $$
>
>  Furthermore, in out-of-distribution (OOD) detection [4, 5], the likelihood ratio between in-distribution and out-of-distribution samples serves as a fundamental decision criterion.
>
> This perspective also offers a theoretical justification for recent empirical practices. For instance, LLaDA 1.5 employs a loss function based on the ratio of conditional likelihoods, interpreted as an alignment score, to supervise model alignment. Our equality result provides a principled foundation for this choice, as it establishes that such likelihood ratios can be reliably estimated from first-order score functions without requiring access to the true data distribution.
>
> Rather than computing two separate likelihood estimates and subtracting them, which can lead to compounded variance, our time-free formulation naturally enables a direct and unified estimator for the log-likelihood ratio, providing improved stability and efficiency:
>
> $$
> \log\frac{p(y^+|x)}{p(y^-|x)} = H\_L \mathbb{E}\_J \left[ \sum_{i\in I\setminus J}\log\frac{p(y^{+,i}|y^{+,J},x)}{p(y^{-,i}|y^{-,J},x)} \right]
> $$
> where $I$ denotes the index set of response tokens and $L=|I|$ The distribution over $J$ follows the sampling scheme described in Corollary 3.10 of our paper.
>
> The effectiveness of our unified estimator is confirmed by our experimental results. We conducted a likelihood ratio estimation experiment on the BeaverTails[9] dataset, where our time-free estimator enables coupled Monte Carlo sampling. This reduces variance substantially compared to standard decoupled sampling. When computing log-likelihood ratios for 500 $(x, y^+, y^-)$ triples, the average variance of MC estimates is significantly lower for our approach:
>
> ### Variance ↓ of Log-Likelihood Ratio Estimation (BeaverTails)
> |#MC  | Coupled       | Decoupled      |
> |-------|--------------|--------------|
> | 5     | 8897.08 | 62469.41 |
> | 10     | 4487.38 | 29107.21 |
> | 15     | 3059.97 | 20695.61 |
> | 20     | 2335.12 | 16514.72 |
>
> These findings highlight an underappreciated advantage of our estimator: it supports efficient and low-variance ratio estimation, which is critical in downstream tasks such as alignment evaluation. We believe that this addresses the reviewer’s concern regarding empirical justification for the advantages of our time-free formulation.
>
> We will update the manuscript to include these new comparisons and clarify the practical advantages of our approach.
>
> > **Q1. Is there anything about your NLL estimator that makes it significantly different in implementation or performance compared to existing methods**
>
> Our NLL estimator differs from existing methods in both formulation and performance. Unlike time-based estimators like RADD, our method uses a time-free formulation over discrete masks, leading to significantly lower variance, as demonstrated in our experiments. Compared to random-order autoregressive models, our estimator is more sample-efficient, requiring fewer forward passes than sequence-length-based permutations. Furthermore, because our estimator is derived from an equality (not a bound), it enables a direct and low-variance estimator for likelihood ratios, which is not possible with standard variational approaches. Experimental results support these advantages.
>
> [1] Shen Nie et al., Large language diffusion models. arXiv 2025.\
> [2] Rafael Rafailov et al., Direct preference optimization: Your language model is secretly a reward model. NeurIPS, 2023.\
> [3] Zhu et al., LLaDA 1.5: Variance-Reduced Preference Optimization for Large Language Diffusion Models, arXiv, 2025.\
> [4] Ren et al., Likelihood ratios for out-of-distribution detection. NeurIPS 2019.\
> [5] Andi Zhang et al., Your Finetuned Large Language Model is Already a Powerful Out-of-distribution Detector. AISTATS, 2025.\
> [6] Rowan Zellers et al., HellaSwag: Can a Machine Really Finish Your Sentence? ACL, 2019.\
> [7] Peter Clark et al., Think you have solved question answering? Try ARC, the AI2 Reasoning Challenge. arXiv, 2018.\
> [8] Yonatan Bisk et al., PIQA: Reasoning about Physical Commonsense in Natural Language. AAAI, 2020.\
> [9] Yixin Nie et al., BeaverTails: Evaluating Language Models by Simulating Human Preference Data. arxiv, 2024.

---

> > ### Comment · Area_Chair_s7hM · 2025-08-05
> >
> > Dear reviewer,
> >
> > The discussion phase is soon coming to and end. It will be great if you could go over the rebuttal and discuss with the authors if you still have outstanding concerns. Thank you for being part of the review process.
> >
> > Regards,
> >
> > Area Chair

---

> ### Comment · Reviewer_pwj1 · 2025-08-05
> **Connections**
>
> Thanks for the thoughtful response, including new results about variance of estimators and discussion of connections to other "time-free" any order autoregressive types of NLL estimators. I will raise my score accordingly.

---

### Official Review · Reviewer_vmxL · 2025-07-09

**Clarity:** 3
**Significance:** 3
**Originality:** 2
**Rating:** 4
**Confidence:** 3

**Summary:**

This paper develops an information-theoretic framework for discrete diffusion models, analogous to the I-MMSE identity in Gaussian settings. Specifically, an I-MDSE identity is derived for discrete diffusion, linking the rate of KL divergence decay to the minimum denoising score-entropy loss. The framework is further extended to masked diffusion models, leading to the I-MDCE relation. Both time-integral and time-free expressions for NLL estimation in conditional and unconditional scenarios are proposed. Experiments on NLL estimation demonstrate the effectiveness of the proposed method.

**Questions:**

The derived I-MDSE and I-MDCE identities link mutual information to score-based losses. What are the practical benefits and how does these relations bring new insights?

**Ethical Concerns:**

["NO or VERY MINOR ethics concerns only"]

**Final Justification:**

I thank the authors for the detailed response. Most of my concerns are addressed and I will raise my rating accordingly.

**Limitations:**

Yes.

**Paper Formatting Concerns:**

I don't find any major formatting issues in this paper.

**Quality:**

3

**Strengths And Weaknesses:**

Strengths:
- The paper is well-structured and the idea of connecting mutual information with score-based losses in discrete diffusion models is interesting. This paper generalizes continuous‑domain I‑MMSE results to discrete domain, offering a new angle to understand discrete diffusion models.
- This paper provides both time-integral and time-free NLL estimators, which are validated by experimental results.

Weaknesses:
- NLL decomposition is achieved via both I-MDSE and I-MDCE relations (Theorem 3.2 and Corollary 3.7), which require the optimal score functions. However, as the optimal score is unavailable, the learned score network is used for approximation, leading to a trivial likelihood estimation via loss function, which is a variational upper bound for log-likelihood. Therefore, the introduction of time-integral NLL estimators seems to be trivial.
- The experimental part lacks comparison. The introduction of the time-free likelihood estimation (Theorem 3.8) is novel. However, the experimental results only show its effectives without comparison to other methods, which is not enough to demonstrate the advantages of time-free estimator as claimed in line 230-232.

---

> ### Author Rebuttal · Authors · 2025-07-31
>
> > **W1. NLL decomposition is achieved via both I-MDSE and I-MDCE relations (Theorem 3.2 and Corollary 3.7), which require the optimal score functions. However, as the optimal score is unavailable, the learned score network is used for approximation, leading to a trivial likelihood estimation via loss function, which is a variational upper bound for log-likelihood. Therefore, the introduction of time-integral NLL estimators seems to be trivial.**
>
>
> We thank the reviewer for raising this important point regarding the role of learned score functions in our NLL decomposition via I-MDSE and I-MDCE.
>
> It is indeed correct that the optimal score function is not directly accessible and must be approximated by a learned model, which introduces a variational aspect into likelihood estimation. However, we respectfully disagree with the characterization that this makes our decomposition or resulting estimators trivial.
>
> The key novelty lies in the fact that our formulation yields an exact equality, not just an upper bound, between the negative log-likelihood and integrals of score-based losses. As noted by reviewer pwj1, this equality result is nontrivial and significant:
>
> *"Even though the result is functionally the same as the variational bound, I like the point that Eq. 11 shows that it ‘fully captures the log-likelihood’. To me, this is quite important as it shows that it is not possible/necessary to tighten the variational bound by outputting higher order moments."*
>
> We believe this exactness is a core strength of our framework. It shows that first-order score functions alone are sufficient to fully characterize the log-likelihood, thereby removing the need for additional assumptions or higher-order statistics often considered in variational estimation. This theoretical clarity also distinguishes our approach from typical score-matching or ELBO-based approximations.
>
> More importantly, this equality is not just of theoretical interest, it enables new practical applications. In particular, we leverage it to construct a unified and low-variance estimator for log-likelihood ratios, as detailed in Section 3. Unlike naive subtraction of two upper bounds (which may not yield a valid or stable estimate), our equality-based formulation supports a direct Monte Carlo estimator with provable correctness and reduced variance, as shown in our experiments. This estimator is particularly useful in tasks like model comparison, importance weighting, and membership inference, and we discuss its variance advantage in our response to W2.
>
> Finally, our equality result offers a principled foundation for empirical practices already in use. For example, LLaDA 1.5 employs a loss function based on the ratio of conditional likelihoods to supervise model alignment. Our formulation formally justifies this practice by showing that such ratios can be reliably estimated from learned first-order score functions, even without access to the true data distribution.
>
> In summary, while our estimator resides within the variational paradigm, the exactness, extensibility, and low-variance properties afforded by the I-MDSE/I-MDCE decompositions represent a nontrivial and practically impactful advancement beyond prior bounds.
>
> > **W2.The experimental part lacks comparison. The introduction of the time-free likelihood estimation (Theorem 3.8) is novel. However, the experimental results only show its effectives without comparison to other methods, which is not enough to demonstrate the advantages of time-free estimator as claimed in line 230-232.**
>
> We thank the reviewer for pointing out the importance of comparative evaluation for the time-free likelihood estimator presented in Theorem 3.8.
>
> We would first like to clarify that a structurally related estimator was independently introduced in the LLaDA paper [1], which we were not fully aware of at the time of submission. While LLaDA focuses on providing an upper bound on the log-likelihood (without proving equality), it reformulates the original RADD estimator as an expectation over the number of masked tokens, a formulation that closely resembles our time-free estimator. As noted in LLaDA:
>
> *"Eq. (12) (RADD formulation) requires over 1000 Monte Carlo estimates for stable results, whereas Eq. (14) (LLaDA’s reformulation) achieves stability with only 128 estimates."*
>
> This observation motivated their reformulation, and we experimentally confirm the same behavior in our setting. Specifically, we conducted a direct comparison between our time-free estimator (Eq. 14) and the time-integral form (Eq. 13) by measuring the Monte Carlo (MC) variance of conditional log-likelihood estimates on 30 sequences with 15 samples each from HellaSwag [6], ARC_hard [7], and PIQA [8] using the LLaDA model. As shown in Table, our time-free estimator consistently achieves lower variance, empirically validating its improved robustness and sample efficiency.
>
> ### Variance ↓ of Estimator (HellaSwag, 30 subset)
> | #MC sample | Time-integral | Time-free |
>  | ----- | ----- | ----- |
> | 128 | 70.9716 | 11.5658 |
> | 256 | 30.1938 | 6.0150 |
> | 512 | 13.3796 | 2.9170 |
> ---
> ### Variance ↓ of Estimator (ARC_hard, 30 subset)
> | #MC sample | Time-integral | Time-free |
>  | ----- | ----- | ----- |
> | 128 | 23.1758 | 5.7318 |
> | 256 | 18.1444 | 2.9612 |
> | 512 | 9.5014 | 1.8200 |
> ---
> ### Variance ↓ of Estimator (Piqa, 30 subset)
> | #MC sample | Time-integral | Time-free |
>  | ----- | ----- | ----- |
> | 128 | 19.76875 | 4.92825 |
> | 256 | 15.14975 | 1.80950 |
> | 512 | 6.50375 | 1.22425 |
>
>
> Additionally, we emphasize a key advantage of our formulation: it provides an explicit distribution over the mask variable $J$, which naturally enables unified (coupled) likelihood ratio estimation, a capability not addressed by LLaDA. Likelihood ratio estimation is central to many downstream tasks in large language models (LLMs). For example:
> - In Direct Preference Optimization (DPO) [2,3], the core loss is based on the log-likelihood ratio between preferred and dispreferred completions: $\log\frac{p^\theta(y^+|x)}{p^\theta(y^-|x)}$
> - In out-of-distribution (OOD) detection [4,5], likelihood ratios between in-distribution and out-of-distribution samples are used as a decision metric.
>
>
> In these settings, prior approaches typically compute two independent likelihood estimates and subtract them, a process prone to compounded variance. In contrast, our time-free estimator enables a direct, unified estimation of the ratio through coupled Monte Carlo sampling:
>
> $$
> \log\frac{p(y^+|x)}{p(y^-|x)} = H\_L \mathbb{E}\_J \left[ \sum_{i\in I\setminus J}\log\frac{p(y^{+,i}|y^{+,J},x)}{p(y^{-,i}|y^{-,J},x)} \right]
> $$
> where $I$ denotes the index set of response tokens and $L=|I|$ The distribution over $J$ follows the sampling scheme described in Corollary 3.10 of our paper.
>
> We validate this through a likelihood ratio estimation experiment on the BeaverTails dataset [9], evaluating 500 $(x, y^+, y^-)$ triples. Our coupled estimator yields substantially lower variance compared to the standard decoupled approach.
>
> ### Variance ↓ of Log-Likelihood Ratio Estimation (BeaverTails, 500 triples)
> |#MC | Coupled       | Decoupled      |
> |-------|--------------|--------------|
> | 5     | 8897.08 | 62469.41 |
> | 10     | 4487.38 | 29107.21 |
> | 15     | 3059.97 | 20695.61 |
> | 20     | 2335.12 | 16514.72 |
>
>
> These findings underscore a practical and underappreciated benefit of our method: it supports efficient, low-variance ratio estimation, which is critical for alignment evaluation and related tasks. In fact, we believe that LLaDA 1.5’s alignment mechanism, which involves estimating conditional likelihood ratios, could be made more efficient by replacing its current two-step estimation with a unified (coupled) VRPO loss, leveraging our formulation.
>
> > **Q1. The derived I-MDSE and I-MDCE identities link mutual information to score-based losses. What are the practical benefits and how does these relations bring new insights?**
>
> The I-MDSE and I-MDCE identities offer a key conceptual insight by establishing an equality, rather than just a bound, between the log-likelihood and integrals of score-based losses. This result demonstrates that first-order score functions are sufficient to fully characterize the likelihood, providing a principled and interpretable foundation for learning in discrete diffusion models.
>
> On the practical side, these identities enable a time-free formulation of both log-likelihood and log-likelihood ratio estimators. Compared to time-integrated approaches like RADD, our formulation yields significantly lower variance, which is especially beneficial in downstream tasks such as alignment evaluation and out-of-distribution detection. Moreover, it supports coupled Monte Carlo estimation of likelihood ratios, improving both stability and efficiency over decoupled approaches.
>
> These advantages are empirically validated in our experiments, as shown in the tables above, where we demonstrate robust performance and reduced variance across multiple tasks and datasets.
>
> [1] Shen Nie et al., Large language diffusion models. arXiv 2025.\
> [2] Rafael Rafailov et al., Direct preference optimization: Your language model is secretly a reward model. NeurIPS, 2023.\
> [3] Zhu et al., LLaDA 1.5: Variance-Reduced Preference Optimization for Large Language Diffusion Models, arXiv, 2025.\
> [4] Ren et al., Likelihood ratios for out-of-distribution detection. NeurIPS 2019.\
> [5] Andi Zhang et al.,. Your Finetuned Large Language Model is Already a Powerful Out-of-distribution Detector. AISTATS, 2025.\
> [6] Rowan Zellers et al., HellaSwag: Can a Machine Really Finish Your Sentence? ACL, 2019.\
> [7] Peter Clark et al., Think you have solved question answering? Try ARC, the AI2 Reasoning Challenge. arXiv, 2018.\
> [8] Yonatan Bisk et al., PIQA: Reasoning about Physical Commonsense in Natural Language. AAAI, 2020.\
> [9] Yixin Nie et al., BeaverTails: Evaluating Language Models by Simulating Human Preference Data. arxiv, 2024.

---

> > ### Comment · Reviewer_vmxL · 2025-08-08
> >
> > I thank the authors for the detailed response. Most of my concerns are addressed and I will raise my rating accordingly.

---

### Note · Authors · 2025-08-15

We sincerely thank the AC and reviewers for the careful reading, constructive feedback, and recognition of our contributions.

>**Main contributions**

Exact NLL Equality: We show that a quantity previously regarded only as a variational bound is exactly equal to the NLL. This has both theoretical and practical value: it tightens the bound and enables likelihood ratio estimation, which is impossible when treated as a bound. As reviewer pwj1 noted, it also removes the need for tightening via higher-order moments.

Time-Free Estimator: Building on this equality, we propose a time-free form with substantially lower variance than the time-integral estimator and supporting unified MC estimation of conditional likelihood ratios. This benefits downstream tasks such as DPO [2], and rebuttal experiments confirm both variance reduction and ratio estimation gains.

Generalization of the I-MMSE Identity: We establish the I-MDSE and I-MDCE identities, paralleling I-MMSE in continuous diffusion. By showing that distribution–loss matching holds between categorical distributions and DSE/DCE losses, we provide a principled justification for these objectives, with our NLL estimators as a first step toward practical use.

>**Regarding Concerns**

Relation to prior work (time-free form): We acknowledge that Eq. (14) in LLaDA [1] presents a prior time-free estimator, but formulated as a variational bound without proof of equality. We rigorously establish the equality, enabling variance reduction and ratio estimation.

Baseline comparisons: We added LLaDA experiments comparing time-integral and time-free forms, confirming the latter’s lower variance and higher ratio estimation accuracy.

Empirical scale: Our main evaluation is on LLaDA, a large-scale LLM. Rebuttal results add variance and likelihood ratio comparisons on the same model, reinforcing the relevance of our findings to realistic large-model settings.

Motivation for I-MDSE/I-MDCE: This framework offers a new theoretical perspective on discrete diffusion analogous to I-MMSE in continuous diffusion, with immediate application to deriving NLL estimators.

We believe that the rigorous theory, practical gains from equality for ratio estimation, and large-scale LLM validation demonstrate the novelty and impact of our work. We thank the reviewers for feedback that improved the paper’s clarity and rigor.


[1] Nie et al., Large language…, arXiv, 2025.
[2] Rafailov et al., Direct Preference Optimization:..., NeurIPS, 2023.

---

### Decision · Program_Chairs · 2025-09-17

**Decision:**

Accept (poster)

**Comment:**

This paper develops an information-theoretic framework for discrete diffusion models, extending I-MMSE results to discrete and masked settings and introducing exact equality-based NLL estimators. The time-free estimator achieves clear variance reduction and enables unified likelihood ratio estimation, which reviewers agreed is theoretically novel and practically useful. Although several reviewers noted the empirical evaluation is limited, all reviewers gave positive scores, and the consensus is that the theoretical contributions and supporting evidence justify acceptance as a poster.